# FIORA: Local neighborhood-based prediction of compound mass spectra from single fragmentation events

Yannek Nowatzky [1,2], Francesco Friedrich Russo[3,4], Jan Lisec [3], Alexander Kister[1], Knut Reinert [2,5], Thilo Muth [2,6] & Philipp Benner [1] ✉

Non-targeted metabolomics holds great promise for advancing precision medicine and biomarker discovery. However, identifying compounds from tandem mass spectra remains a challenging task due to the incomplete nature of spectral reference libraries. Augmenting these libraries with simulated mass spectra can provide the necessary references to resolve unmatched spectra, but generating high-quality data is difficult. In this study, we present FIORA, an open-source graph neural network designed to simulate tandem mass spectra. Our main contribution lies in utilizing the molecular neighborhood of bonds to learn breaking patterns and derive fragment ion probabilities. FIORA not only surpasses state-of-the-art fragmentation algorithms, ICEBERG and CFM-ID, in prediction quality, but also facilitates the prediction of additional features, such as retention time and collision cross section. Utilizing GPU acceleration, FIORA enables rapid validation of putative compound annotations and large-scale expansion of spectral reference libraries with high-quality predictions.

Progress in non-targeted metabolomics is limited by the scarcity of high-quality reference spectra. This discipline promotes an unbiased exploration of metabolites within biological systems and is facilitated by liquid chromatography-mass spectrometry (LC-MS)[1]. In high-throughput settings, compounds are ionized and isolated based on their biophysical properties and ion mass, and then fragmented into product ions[2,3]. The product ions are recorded as peaks in a tandem mass spectrum (MS/MS) and act as a signature or fingerprint for a given molecule. However, in 2015, da Silva et al. showed that only a small fraction of MS/MS spectra from non-targeted experiments can be annotated by searching spectral libraries of reference standards due to their incomplete nature[4]. They coined the term "dark matter" to describe the overwhelming number of unidentified signals and chemical species that remain unknown.

In the past decade, this situation has led to the development of various algorithms that attempt to infer compound identity directly from mass spectra, so-called in silico methods. These include, but are not limited to, CSI:FingerID as part of the SIRIUS suite[5,6], MS-FINDER[7,8], and MS2LDA[9]. Despite these advancements, the identification rates of "unknown" compounds remain low. This was demonstrated in the 2016 CASMI challenge, where in silico methods achieved a recall rate of only up to 34%, when annotating spectra of previously unknown compounds[10]. Meanwhile, in the most recent CASMI challenge in 2022, identification rates were below 30%[11].

To improve compound identification by amending existing libraries, many research groups attempt to build theoretical product ion spectra from molecular structures[12]. These spectra serve as reference and allow the expansion of public spectral libraries when experimental metabolite spectra are unavailable. In particular, in silico fragmentation algorithms simulate the MS fragmentation process, and exploit the chemical and structural properties of the molecules to predict fragment ions or infer their identity. This process is facilitated

[1]Section VP.1 eScience, Federal Institute for Materials Research and Testing (BAM), Berlin, Germany. [2]Department of Mathematics and Computer Science, Freie Universität Berlin, Berlin, Germany. [3]Department of Analytical Chemistry and Reference Materials, Organic Trace Analysis and Food Analysis, Federal Institute for Materials Research and Testing (BAM), Berlin, Germany. [4]Institute of Pharmacy, Freie Universität Berlin, Berlin, Germany. [5]Department of Computational Molecular Biology, Max Planck Institute for Molecular Genetics, Berlin, Germany. [6]Data Competence Center MF 2, Robert Koch Institute, Berlin, Germany. ✉e-mail: philipp.benner@bam.de

by large knowledge bases that provide known chemical structures and properties, such as PubChem[13] and HMDB[14]. These are orders of magnitudes larger than spectral databases[15], such as GNPS[16] and METLIN[17]. Nonetheless, the accurate prediction of MS/MS spectra remains a significant challenge due to the scarcity of high-quality training data and algorithms must be thoroughly evaluated to determine their effectiveness for previously unreferenced or unseen metabolites.

At the same time, accurate annotation of compound spectra is paramount to metabolomics. Non-targeted screening approaches have become increasingly popular in clinical diagnostics, drug response monitoring, and the characterization of intracellular molecular mechanisms[18,19]. The link between the metabolome, which describes the biochemical phenotype, and the genotype, microbiome, and environmental exposures in human health and disease makes metabolomics an invaluable tool for biomarker discovery and hypothesis generation[20]. This places particular emphasis on the development of compound identification methods that must be highly efficient and accurate. Ongoing advances in in silico fragmentation methods can deepen our understanding of MS-based compound fragmentation, expand the search space for spectral libraries, and offer additional levels of confidence to other identification methods.

Bond dissociation is a key concept behind compound fragmentation, as covalent bonds are cleaved during MS/MS, producing fragment ions that appear in the mass spectrum[12,21]. Typically, one fragment is lost, referred to as neutral loss, while the fragment on the other side of the fragmentation site retains the charge and is observed as a peak in the m/z dimension. Multiple bond cleavages and hydrogen rearrangements may occur. The abundance of the fragment ions and therefore the probabilities of the corresponding bond breaks are directly tied to the peak intensity. In silico fragmentation algorithms attempt to identify breakpoints in the molecular structures and use these to impute ion probabilities and peak intensities. The output is a simulated mass spectrum. In addition, structural fragment annotations and fragmentation pathways can be retained. Figure 1 illustrates a comparison of the experimental and computational fragmentation workflows.

CFM-ID is an advanced machine learning (ML) algorithm that predicts transition probabilities between fragments. Since its introduction by Allen et al. in 2015[22] it has undergone many improvements, with the latest version 4.0 being published in 2021[23]. CFM-ID is widely regarded as a pioneer in ML-driven in silico fragmentation of molecular structures. The method models the fragmentation process as a stochastic, homogeneous Markov process and learns model parameters using an expectation-maximization algorithm[23]. However, CFM-ID suffers from slow training and prediction performance, often rendering it insufficient for predicting a large candidate space of possible structures or rescoring many tentative identifications. Despite its complexity, CFM-ID does not rely on modern deep learning structures like graph neural networks.

Graph neural networks (GNNs) have become popular over the past decades, in part due to applications in cheminformatics and drug discovery[24]. Their ability to learn and characterize molecular structure graphs has proven to be essential for predicting molecular properties, such as solvation free energy or metabolic stability. While GNNs have recently gained attention in metabolomics applications, their full potential still remains untapped.

One popular approach involves using GNNs to embed the molecular structure and then predict vectors of fixed length representing binned MS/MS spectra. Zhu et al. (2020)[25] attempted this using graph convolutional neural networks (GCNs) and graph attention networks (GATs), while Young et al. (2021)[26] used a graph transformers architecture[27] for spectra binned at 1 Da resolution. In 2023, Park et al.[28] introduced a GNN combining the molecular structure graph with a heterogeneous motif graph, and the QC-GN²oMS² model by Overstreet et al. (2023)[29] adds quantum chemical bond features to improve spectrum prediction at high resolution. While using modern deep learning architectures, none of these methods leverage the molecular graph structure to their full potential. All the approaches compute a single embedding from the input graph (pooled together from the

**Fig. 1 | Illustration of experimental MS/MS fragmentation in comparison to the in silico fragmentation workflow.** The bottom panel illustrates the computational workflow of fragmenting the molecular graph structure. Fragment ions are predicted as charged substructures with the addition and subtraction of hydrogen atoms, closely simulating the experimental spectrum (on top).

node features), which is then used to predict a fixed-length vector representing the mass spectrum. This makes the assumption that the models are able to learn all fragment ions from a singular graph representation and directly associate them with the correct m/z bin. In doing so, crucial information about learned subgraph structures around the breaking points becomes entangled when node features are pooled together, making it much harder to learn local properties. This information is accessible at the bond cleavage sites, but requires more complex and elaborate model structures than binned peak prediction. In addition, the fixed output format limits the models to a specific mass resolution, required for binning the spectra. High-resolution predictions get increasingly harder and more training data is required to increase mass accuracy as technology advances. In contrast, fragmentation algorithms that iterate and break chemical bonds allow the direct prediction of fragment ions and the calculation of the exact peak position, so that mass resolution is infinite. Such fragmentation methods are therefore timeless and remain unaffected by the next technological leap in mass specificity.

In 2023, Murphy et al. introduced the graph network GRAFF-MS that predicts molecular formulas of fragment ions and neutral losses from a fixed vocabulary, bypassing the mass resolution problem altogether[30]. They make a case against bond breaking, as most peak signals can be explained by a fixed vocabulary of sufficient length. However, there are several advantages to using bond dissociation for fragment inference. For one, explicitly modeling bond breaks retains a higher level of explainability and allows the annotation of fragments (and fragmentation pathways), which is essential to understanding and validating MS/MS. As all theoretical breaking points are covered, this approach is able to find previously undiscovered fragments in unknown chemical species that may be missing from a fixed vocabulary. Lastly, GRAFF-MS relies on full graph embeddings for molecular formula prediction and, similar to the binned peak prediction models, does not fully exploit the graph substructures around breaking points.

ICEBERG[31] and SCARF[32] are two spectral prediction methods recently developed by Goldman et al. (2024) that strike a balance between physically-grounded fragmentation algorithms and the advantages of "black box" peak intensity predictors using deep neural networks. Both models have two separate modules that work in conjunction. The first module generates potential fragments (or molecular formulae in the case of SCARF) and the second module predicts intensities for the set of fragments using Set Transformers[33]. Although these models provide some explanation of possible fragmentation events and predict peaks with high accuracy, they do not take into account the features of the broken bonds or a local representation of their surrounding molecular neighborhood when predicting fragment intensities. We argue that these factors are the most important criteria in determining break probabilities and hydrogen rearrangements. Interestingly, ICEBERG uses GNNs to embed the molecular structure and goes as far as modeling fragmentation events through the stepwise removal of atoms. However, the final intensity prediction trivializes the fragmentation events (and adherent substructures) to the fragment and parent molecule embeddings and the number of bonds removed. Both, ICEBERG and SCARF do not consider covariates, such as collision energy, which is a major factor influencing fragmentation events and, consequently, the resulting fragment ion abundances. Both models operate only in positive ion mode.

We present FIORA, a modular graph network structure that stands for **F**ragment **I**on **R**econstruction **A**lgorithm. FIORA is a multi-purpose framework designed to predict various spectral features. What sets FIORA apart is the commitment to expressing each bond cleavage with its local molecular neighborhood. This marks a departure from the typical approach of predicting MS/MS spectra or complete sets of fragments based on a summarized representation (embedding) of the molecule as seen in many recent algorithms. Instead, FIORA evaluates bond dissociation events independently, on the basis of their surrounding molecular structure, thereby simulating the physical fragmentation process of MS more directly. FIORA uses state-of-the-art GNN architectures and formalizes fragment ion prediction as an edge-level prediction task within the molecular structure graph. In doing so, FIORA makes great use of high-performance GPUs and has a strong emphasis on explainability in its decision-making process, but is so far limited to single-step fragmentation. FIORA reconstructs complete MS/MS spectra for both positive and negative ionization modes ([M+H]+ and [M-H]- precursors). In addition, FIORA estimates retention times (RT) and collision cross sections (CCS), which add further dimensions for MS-based compound identification and is a truly original addition to spectral prediction software. We benchmark the performance against the top-performing methods, CFM-ID and ICE-BERG. Our results demonstrate that FIORA learns fragmentation patterns relatively independent of the structural similarity between the training set and unknown compounds. This ensures a high degree of generalizability for modeling truly unknown structures and sets the state of the art for spectral feature prediction. FIORA is open source (MIT license) and freely available on GitHub at https://github.com/BAMeScience/fiora[34].

## Results
### Overview of the fragmentation method
The core idea behind FIORA is to predict mass spectra indirectly by anticipating molecular bond breaks that occur during the fragmentation process of tandem MS. To this end, we employ a GNN to learn hidden representations of the molecules and formulate bond breaks as an edge property prediction task. Fragment ions (and neutral losses) are modeled as a direct consequence of edge removal from the molecular graph. Our model takes into account the local neighborhood of each bond, thus exploiting a close-to-complete chemical representation relevant for deciphering fragmentation events and ion rearrangements.

Subsequently, FIORA models MS/MS signals as a probability distribution over the predicted fragment products following a single bond dissociation. It builds upon the statistics of independent break tendency values introduced by Allen et al.[22]. We extend this concept to directly estimate fragment ion probabilities featuring multiple hydrogen rearrangements, as well as estimate the precursor probability. Figure 1 illustrates the in silico spectral prediction workflow. Further details on the model and the spectral reconstruction algorithm can be found in the "Methods" section.

The graph network module also allows traditional molecular property prediction, as seen in other fields such as drug property prediction[24]. We utilize this for learning RT and CCS values with neural network submodules using the molecular graph embeddings that are a result of the fragmentation process. In this way, FIORA provides multiple MS/MS feature dimensions to match experimental data, which can be used to improve compound identification. To the best of our knowledge, FIORA is the only model that simulates complete MS/MS compound spectra, including fragment annotation, RT, and CCS values. Furthermore, FIORA is designed to be flexible towards various experimental setups and includes covariate features, such as ionization mode, a continuous scale of collision energies, and compatibility with many types of MS instruments.

### Data
The data in this study have been aggregated from three spectral libraries (NIST'17, MS-DIAL, and MSnLib)[35–37] and two CASMI challenges (2016 and 2022)[10,11]. It should be noted that, in comparison with analogous studies (e.g., Goldman et al.[31]), an earlier version of NIST was used. While NIST'17 is more limited in terms of compound content and as such, the amount of training data it provides (compared to NIST'20), it allowed us to retrain the competing algorithms in a fair manner (given the comparably sized dataset used to train CFM-ID).

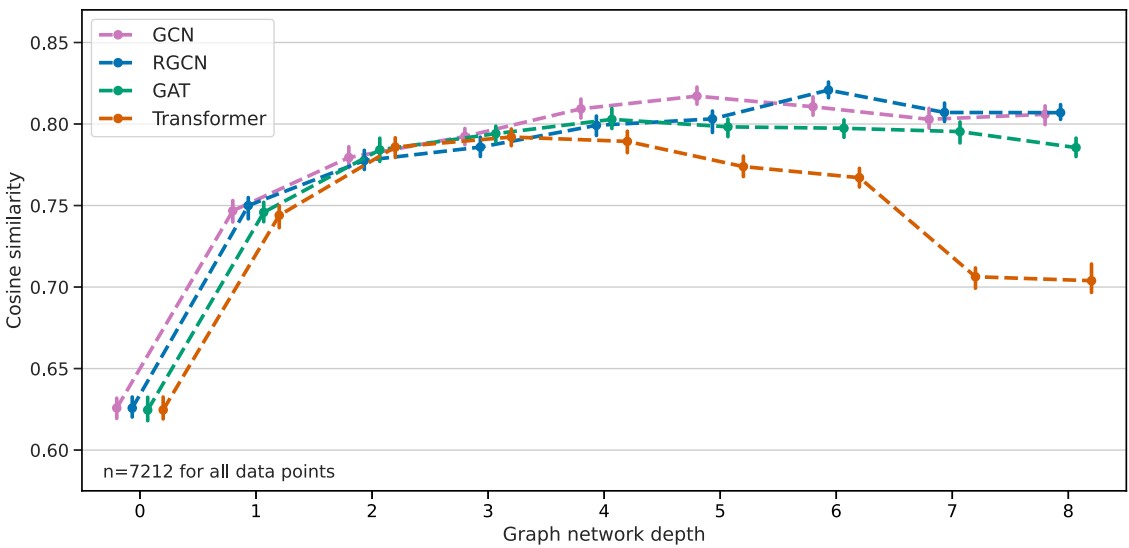

**Fig. 2 | Spectral prediction performance for various GNN architectures.** The median cosine similarity is evaluated on the validation split, with error bars denoting the 95% confidence intervals. Note that the interquartile ranges of the data are considerably broader, as shown in Supplementary Fig. 24. Cosine similarity reaches its peak between a network depth of 4 and 6 layers before it falls off again. The graph convolution (GCN) and relational graph convolutional networks (RGCN) perform better than the attention based mechanisms (GAT and Transformer).

Furthermore, by assembling a diverse collection from multiple sources, we are able to cover a broad range of compound classes, instruments, and experimental conditions. This, in turn, allows for the training and prediction of additional MS features, such as RT and CCS values. Through rigorous testing under various MS/MS conditions, presented in the subsequent sections, we ensured our evaluations are state-of-the-art, albeit less conventional. All spectral libraries have been meticulously curated for spectral quality and metadata, particularly with respect to collision energies, as described in detail in the "Data preparation" section. Our emphasis on predominantly open access data enables the method to be freely available and reproducible.

Training and test data are split as follows: for the default FIORA model, the NIST'17 and MS-DIAL libraries have been merged, with an 80% split allocated to training and 20% to validation and testing. To facilitate public use of the software, we have also trained FIORA on 80% of the MSnLib library and made the model weights available on GitHub[34]. This open-sourced version of the model is referred to as FIORA-OS from this point forward. The remaining 20% are again divided between a validation set and an additional (independent) test set for all models. Moreover, we assess spectral prediction performances for the CASMI challenges in 2016 and 2022. As part of the challenges, the experimental MS/MS data were used to test in silico algorithms in their ability to identify compounds that were not recorded in spectral libraries. Although spectral reference libraries have since been expanded, the CASMI challenge compounds remain important cornerstones for cross-referencing the performance of metabolomics software and have been used for benchmarking purposes in many studies[23,31]. All compounds from the test sets were explicitly excluded from the training process. For further details refer to the "Training and testing" section.

### Model selection
FIORA shows versatility by not being constrained to a single model architecture. Its modular design allows for multiple prediction targets and effortless integration of different deep learning architectures, as will be evaluated in this section. Note that while FIORA learns all prediction targets, i.e., fragment intensities, RT, and CCS values, our primary focus is on the spectral predictions in terms of hyperparameter tuning and model selection.

The initial model selection is conducted on the validation split, examining the performance of popular graph network architectures with variable network depths. The effect of different architectures on cosine similarity (using square root intensities, as described in "Evaluation metrics") between the spectral reconstruction and the ground truth validation spectra is shown in Fig. 2. All other hyperparameters are fixed. Interestingly, the graph convolutional network (GCN) and relational graph convolutional network (RGCN) outperform attention-based networks, i.e., GATs and Transformers, especially as the network depth increases. There is a sweet spot in graph depth at 4 to 6 layers, which maximizes cosine similarity to the validation set. Notably, 0 or 1 graph layers are significantly less powerful because very little structure information is aggregated. Similarly, a high number of graph layers (>7) leads to reduced performance. GNNs are known to lose expressive power when too many graph convolutions are applied, as the hidden node representations become indistinguishable[38]. This is particularly evident for the attention-based mechanisms, suggesting that they may be less effective at predicting fragment ions. The integration of bond type information to graph convolutions, as seen in the RGCN compared to the GCN, appears to have a small positive impact on prediction quality for high network depth.

Note that fragment ion prediction is centered around bond breaks. This process incorporates bond (edge) features, node embeddings of the two neighboring atoms and covariate features, such as collision energy, at the final layers. This way, at depth 0 the predictor is aware of the bond type and the connected atoms. Similarly, at depth 5, substructure information of up to 6 atoms from either side is aggregated, thereby covering a complete 6-cycle ring structure. The RGCN with a depth of 6, which performed best on the validation set, was selected for subsequent benchmarking on the four test sets. Exact model specifications can be found in the "Methods" section and the Supplementary Information.

### Spectral prediction quality
Of the algorithms discussed in the "Introduction", only CFM-ID and ICEBERG are in silico fragmentation algorithms in a true sense, i.e., methods annotating fragment structure and modeling break events. Recently, Goldman et al.[31] conducted a comprehensive benchmarking study, in which ICEBERG outperformed all other spectral prediction software. For this reason, we compare FIORA with ICEBERG and CFM-ID.

**Table 1 | Median cosine similarity of spectral predictions to ground truth test spectra**

| Unique compounds | Test split + 895 | Test split - 437 | MSnLib + 785 | MSnLib - 492 | CASMI 16 + 381 | CASMI 16 - 139 | CASMI 22 + 160 | CASMI 22 - 98 |
|---|---|---|---|---|---|---|---|---|
| FIORA | **0.81** | **0.79** | **0.65** | **0.61** | **0.77** | **0.77** | 0.29 | **0.32** |
| CFM-ID | 0.67 | 0.57 | 0.48 | 0.41 | 0.70 | 0.59 | **0.38** | 0.29 |
| ICEBERG | 0.72 | - | 0.58 | - | 0.71 | - | 0.36 | - |
| FIORA-OS | 0.69 | 0.73 | 0.70† | 0.68† | 0.72 | 0.78† | 0.27 | 0.31 |

The columns are arranged according to the test sets and precursor ion modes (positive and negative), with the best-performing model highlighted in bold. ICEBERG operates in positive ionization mode only. For more information on the test sets, please refer to the "Training and testing" section. The FIORA open source (OS) version is separated from the direct comparison. The † symbol indicates whenever FIORA-OS outperforms all other software.

Table 1 shows the overall cosine scores, separated for positive [M+H]+ and negative [M-H]- precursor charges. Note that the precursor peak is included in this statistic, thereby reflecting the model performance over the full spectrum. The role of the precursor peak concerning cosine similarity is subsequently examined, as its exclusion from the MS/MS spectrum may provide a more accurate reflection of a model's practical performance in identifying structure-specific peaks. Other types of precursors or adducts, e.g., [M]+ and [M + Na]+ ions, are currently not supported by the default FIORA model. ICEBERG was retrained on the exact same dataset as we trained FIORA on (for positive spectra exclusively). For CFM-ID this was not feasible and the latest model, pre-trained on the METLIN library[17], was used instead.

FIORA's predicted MS/MS spectra exhibit the highest median cosine similarity to reference test spectra for the test split, MSnLib, and CASMI 16 datasets, with a gain ranging from 10% to 49% over the runner-up. The relative improvements are more pronounced for the negative test sets. For the CASMI 22 dataset, the overall cosine scores are significantly lower for all algorithms compared to the other test sets. FIORA is slightly better than CFM-ID in the negative ionization mode, but falls short in the positive mode. The performance between ICEBERG and CFM-ID is similar, with ICEBERG being slightly better on the test split, MSnLib, and CASMI 16, but CFM-ID being superior on the CASMI 22 dataset.

Interestingly, the low cosine scores for CASMI 22 coincide with the results reported by Goldman et al.[31]. They postulate that CASMI 22 may be a more challenging dataset due to a high degree of structurally distinct compounds. However, this is also the case for the MSnLib data, as detailed in the Supplementary Information. While the cosine scores for MSnLib are overall lower (by ~20% compared to the test split), the difference is much smaller than for CASMI 22. Furthermore, we identified several spectra that challenge the quality of CASMI 22. A section in the Supplementary Information is specifically dedicated to explaining the discrepancies with the CASMI 22 dataset, including a more nuanced performance analysis. Based on these findings, we conclude that the dataset cannot be considered canonical, but decide to report the results for the sake of completeness.

The performance of the FIORA-OS model, which was trained on MSnLib data, is comparable to that of the default FIORA model. However, there are notable differences between the two. FIORA-OS is more accurate on the MSnLib test sets, but less so on the NIST/MS-DIAL test splits. This discrepancy can be attributed to the different compound and covariate compositions observed during the training phases. Curiously, FIORA-OS predicts negative [M-H]- spectra with markedly greater accuracy than positive ones. For instance, FIORA-OS achieves the highest median cosine similarity of all tools on the CASMI 16 negative set, but exhibits a decreased performance for positive CASMI 16 data. This trend is even more pronounced when considering non-precursor metrics.

By excluding the precursor peak, the cosine similarity score describes spectral prediction quality exclusively based on fragment peaks. This is of particular significance as it demonstrates the degree to which each software is capable of differentiating between compounds with the same precursor mass. Following the removal of the precursor,

we observe a notable decline in cosine similarity for all tools. On average, FIORA exhibits a reduction in cosine similarity of 0.06 per spectrum. CFM-ID exhibits an average loss of 0.08. In general, the decline in performance is more pronounced for negative spectra, particularly in the context of CASMI 2016, where the similarity scores decreased significantly by 0.26 on average for both CFM-ID and FIORA. This indicates that the accuracy of [M-H]- predictions is contingent upon the precise estimation of precursor intensity, and that negative fragment ions are considerably more challenging to predict. The precursor accounts for 25% of the total ion count in negative test spectra, compared to only 21% in positive mode. As a result, CFM-ID attains a median cosine similarity for CASMI 16 negative spectra as low as 0.14, while FIORA sits at 0.44. The median scores per dataset are presented in Supplementary Table 1. It should be noted that the average change per spectrum is not equivalent to the median performance. ICEBERG is the least affected by the change, with an average loss of 0.03 compared to 0.05 by FIORA on positive data. Without the precursor, ICEBERG performs markedly better than CFM-ID across all positive datasets. However, FIORA continues to outperform all other software on all datasets except CASMI 22, although the margin by which it surpasses ICEBERG is significantly smaller. It can be concluded that a portion of FIORA's advantage–and to a degree that of CFM-ID–is due to the accurate prediction of precursor intensity. FIORA considers continuous collision energies (ranging from 0 to 100 eV) as an input feature, which are key factors in the degradation of the precursor molecule. An extensive analysis of the impact of collision energies on MS/MS fragmentation is found in the Supplementary Information, reinforcing that collision energies are indeed crucial covariates deserving careful consideration.

Furthermore, we could establish a correlation between the distribution of cosine similarity and an enhanced compound identification performance of FIORA. This was tested by ranking a series of compound candidates based on spectral predictions, as outlined in the Supplementary Information. While the non-precursor cosine score is better suited to distinguishing the correct compounds from false candidates, the results indicate that integrating the precursor intensity to a small degree may further improve compound retrieval rates. Various additional statistics and figures on cosine similarity, compound, and MS/MS distributions can be found in the Supplementary Information.

In light of these findings, it is evident that FIORA exhibits superior performance in the majority of test cases. However, some notable distinctions are worth mentioning. FIORA is highly proficient in estimating precursor intensity across a wide array of collision energies. Since precursor intensity is inextricably linked to molecular stability, it remains an important descriptor of the metabolites. On a dataset such as CASMI 2016, which is composed of stepped spectra (measured over three steps of increasing collision energies), the improvements over ICEBERG are minor. ICEBERG essentially predicts average spectra, and the influence of variable collision energies is diminished in this context. Moreover, the improvements brought about by FIORA's fragmentation algorithm are more pronounced for negative mode spectra. This could be attributed to the fact that positive and negative spectra

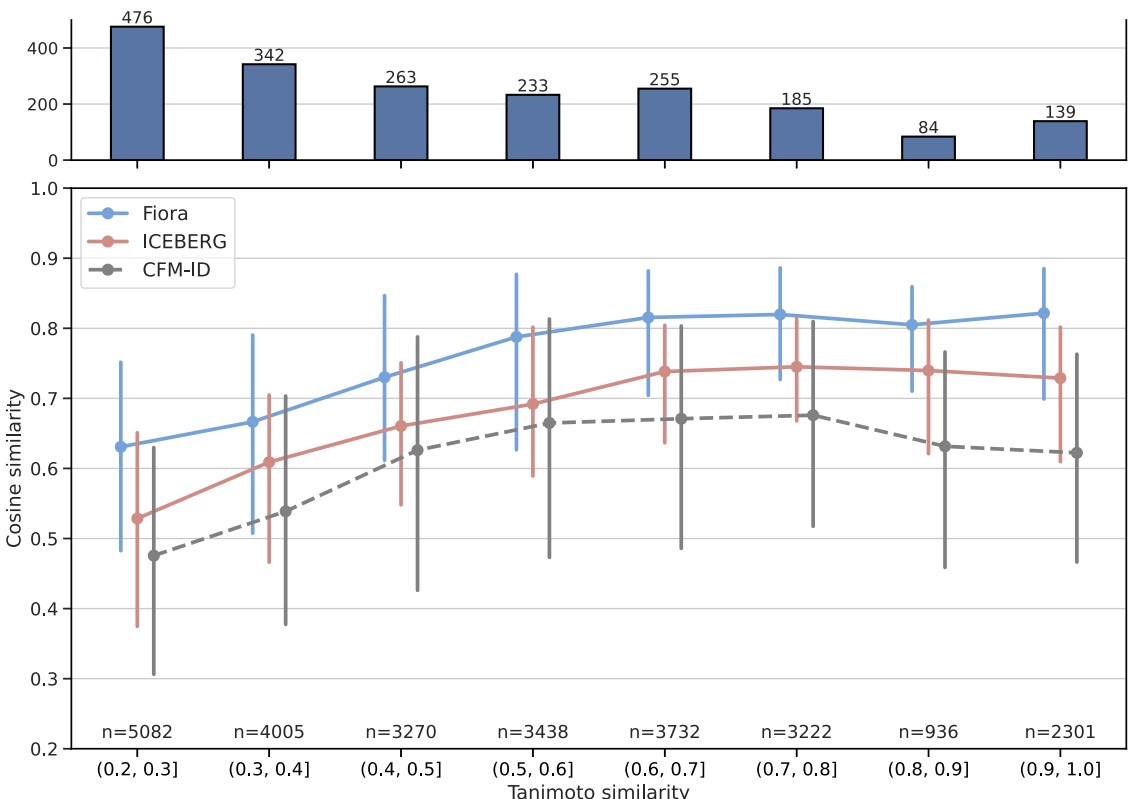

**Fig. 3 | Cosine similarity at intervals of structural similarity of test compounds to training compounds.** Structural similarity is measured by the maximum Tanimoto similarity (Jaccard index) using Morgan fingerprints with 2048 bits and a radius of 3. The data points present the median cosine similarity and error bars denote the interquartile ranges. On the top panel, the number of unique compounds for each group is shown. Results are shown for positive ionization spectra to ensure the same dataset for all algorithms. Since CFM-ID was trained on a different dataset, the intervals do not reflect the actual Tanimoto similarity for the model. CFM-ID's cosine scores still provide an overview of the prediction performance for each interval, evaluated with a more independent model.

were trained simultaneously with the same model. In contrast, CFM-ID employs separate models trained for either positive or negative spectra.

## FIORA generalizes well across compound classes and to unknown compounds

Fragmentation algorithms, unlike spectral prediction models, have the ability to explain their output by virtue of their algorithmic design. Specifically, predicted peaks are annotated by fragment ions (and potentially fragmentation pathways), grounding the predictions in a relatable and physics-inspired process. However, due to the complexity of deep learning models with millions of parameters, it is often unclear how models generate their output. It is important to determine whether models take shortcuts or overfit to specific input types or features. In this section, we provide an overview of FIORA's ability to generalize to unknown compounds, evaluate the performance across different compound classes, and contextualize the latent feature representation acquired by FIORA's graph module with the structural properties of the compounds under study.

To assess the extent to which the fragmentation methods can generalize to uncommon structures, we compute the Tanimoto similarity (Jaccard index) between all test and training compounds based on their Morgan fingerprints (2048 bits; radius 3). The maximum Tanimoto similarity for each test compound serves as a meaningful descriptor of its structural similarity to the entire training set. Compounds with low Tanimoto similarity are arguably more difficult to model and represent a distribution of unreferenced compounds that are dissimilar to those in the spectral libraries. These compounds are of particular interest for spectral prediction because they cannot be easily related to other reference compounds using methods such as

Molecular Networking, and constitute the unexplored chemical space, i.e., metabolomic "dark matter"[4]. Note that that for each compound, multiple (test) spectra may be recorded and evaluated. In order to clearly distinguish between different molecular structures, unique compound numbers are reported. Figure 3 depicts the median cosine similarity at different levels of Tanimoto similarity. Interestingly, for compounds of medium to high structural similarities to the training set (Tanimoto scores between 0.6 and 1), FIORA's prediction quality remains stable, with a median cosine similarity of 0.8 and above. The prediction performance declines linearly for Tanimoto similarity values below 0.6. For the most dissimilar set of compounds (Tanimoto similarities of 0.2 to 0.3), the median cosine similarity is 0.63, indicating that FIORA's performance remains robust when generalizing to unfamiliar structures. The curve depicting ICEBERG's performance is very similar to that of FIORA, but with an overall lower cosine similarity. ICEBERG appears to have more difficulties predicting spectra for compounds with a very low Tanimoto similarity of 0.2 to 0.3. CFM-ID was pre-trained on a different dataset, so the intervals do not correctly reflect the Tanimoto similarity between training and test compounds. This is evident in Fig. 3, where there is a lack of a clear upward trend for CFM-ID and wider interquartile ranges. However, CFM-ID takes on the role of a (more) independent evaluator of the different subsets of compounds. Importantly, the performance of CFM-ID is also lower for low Tanimoto similarities between 0.2 and 0.4, indicating that these compounds are either in fact rather uncommon or at the very least challenging to predict. We conclude that FIORA generalizes quite well to structurally dissimilar compounds, but shows a visible drop in quality for compounds with Tanimoto scores below 0.6, as expected. Still, the relative loss is not higher than that of ICEBERG, and the apparent decline in performance illustrates the inherent difficulty of

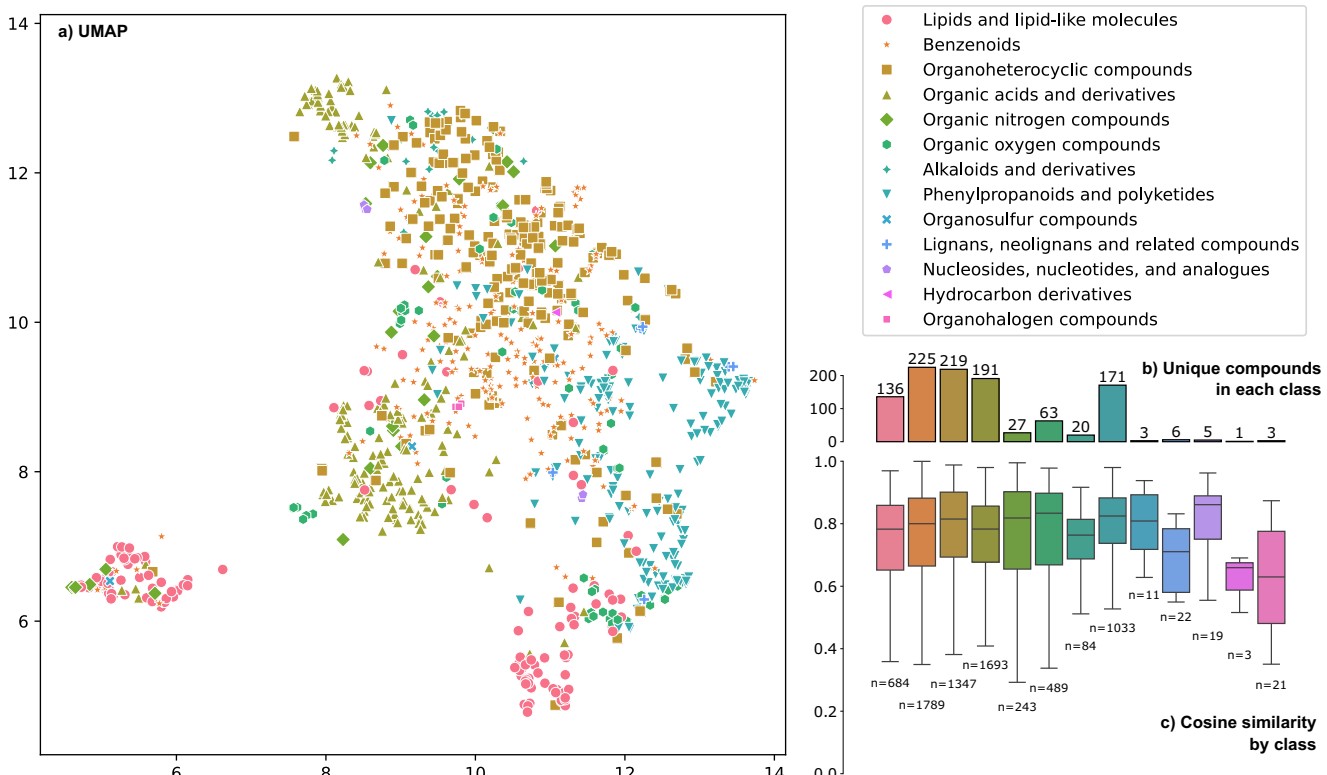

**Fig. 4 | UMAP visualization and distribution analysis of compound superclasses. a** Uniform Manifold Approximation and Projection (UMAP)[40] visualization of the molecular graph embeddings. Each point corresponds to a unique compound and is color-coded based on compound superclasses, annotated by ClassyFire[39]. Clustering of compounds according to their superclasses is evident, indicating the learned structural relationships by FIORA during training. **b** Number of unique compounds per class. Each compound can have several spectra in the test set. **c** Cosine similarity between experimental and predicted test spectra split according to the compound superclasses. The boxes represent the interquartile ranges, with the median at the centre, and the whiskers extend to the limits of the distribution (excluding outliers). Prediction performance is consistent across most superclasses, with median scores of above 0.7.

predicting MS/MS spectra of truly unknown structures. The same statistic for non-precursor spectra can be found in the Supplementary Information, showing an overall similar picture. All cosine scores are slightly lower and improvements of FIORA over ICEBERG are less pronounced, but clearly visible for all sets of compounds.

Moreover, the prediction quality of FIORA is very stable across different compound classes. Figure 4c shows the cosine similarity scores for individual compound superclasses, annotated using ClassyFire[39]. FIORA consistently achieves a median cosine score well above 0.7 for all compound superclasses, except for *Organohalogen compounds* and *Hydrocarbon derivatives*, which have median scores 0.63 and 0.66, respectively. However, there are only very few representative compounds in these superclasses, a single one (three spectra) in the case of *Hydrocarbon derivatives* (see Fig. 4b), so this result carries little statistical weight. Similarly, *Nucleosides, nucleotides and analogues* have a very high median cosine similarity of 0.86 based on only 5 unique compounds (19 spectra). Overall, FIORA's performance appears to be robust across the test set without emphasizing specific compounds at the superclass level.

At the same time, the shared molecular structures within the compound superclasses have a significant impact on the latent representation that FIORA learns. Figure 4a shows FIORA's graph embeddings (mean pooled over all nodes) after a UMAP dimensionality reduction to 2-D[40], with each compound colored according to its corresponding superclass. Keep in mind that FIORA is not trained to produce a meaningful compound representation that can be used for property prediction, but rather to solve the edge break problem using local neighborhoods. Graph layers are not affected by the training of RT and CCS values, making the global molecular structure embedding

purely a by-product of the fragmentation method. We observe that the compound graph embeddings form structural clusters, highlighted by the superclass annotation in Fig. 4a. This clustering effect goes beyond the superclass level. For instance, *Lipids and lipid-like molecules* are grouped together in two main clusters (seen at the bottom left and bottom right of the UMAP). Upon closer examination, one cluster is dominated by *Glycerophospholipids*, *Fatty Acyls*, and *Sphingolipids*, while the other cluster contains *Prenol lipids*, *Steroids*, and *Saccarolipids*. These lipid classes also separate well within each cluster, which is shown in the Supplementary Information.

Naturally, the clusters are also the result of similarities in the molecular graphs that FIORA receives as input. These may have similar element compositions or share certain structural elements within the superclasses. Nevertheless, it highlights the expressive power of GNNs in the context of molecular structures and establishes that local property prediction can still generate meaningful global embeddings.

Furthermore, we could demonstrate considerable improvements in the prediction of RT and CCS values using this graph embedding, which is presented in the "Retention time and collision cross section" section. FIORA not only produces structurally meaningful embeddings, but also encapsulates critical information about the 3-D structure (CCS) and chromatographic properties (RT), and quite possibly other pertinent molecular properties as well.

## Retention time and collision cross section

FIORA's architecture was designed to support additional prediction targets through individual submodules branching off the graph convolutional layers. As a proof of concept, we show that FIORA can accurately predict RT and CCS values. The model was trained on a

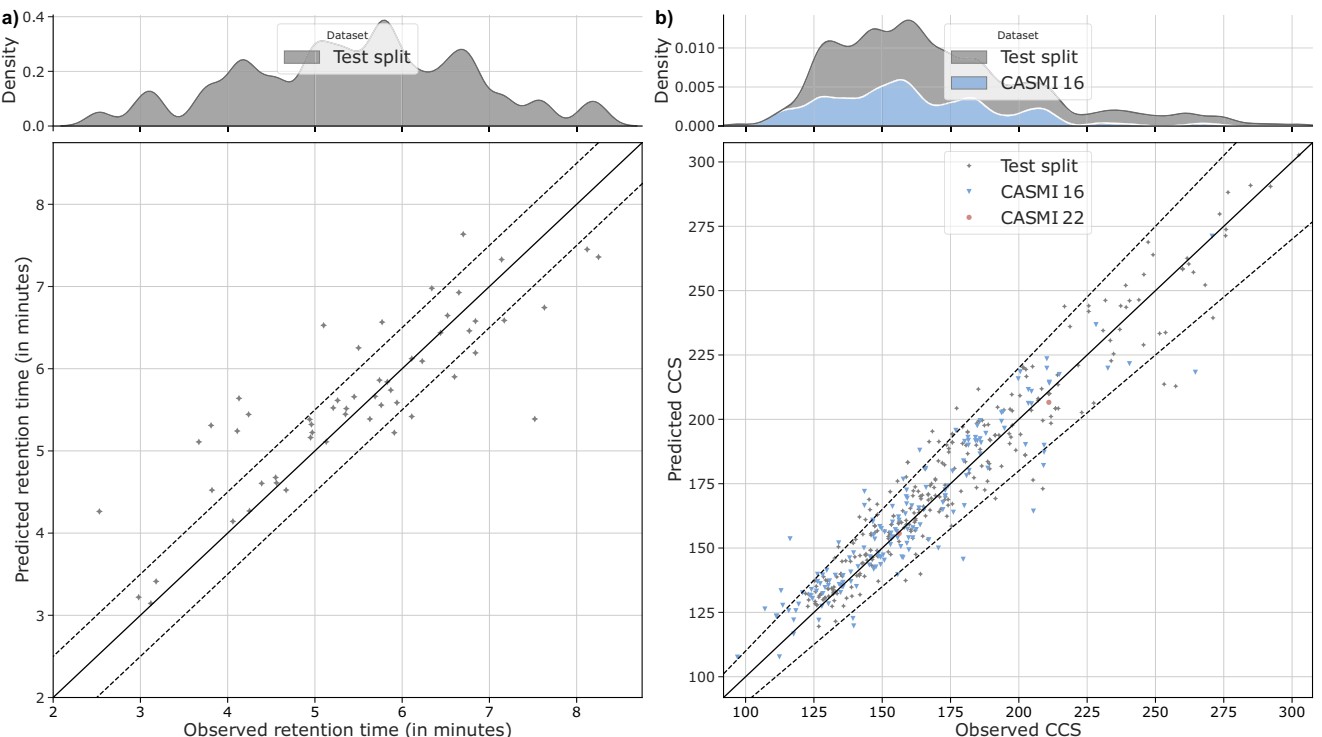

**Fig. 5 | Parity plot of RT and CCS predictions by FIORA. a** Predicted RT values are presented in comparison to observed RT values (ground truth). The RT values for the test sets were retrieved from the BMDMS-NP library[57]. The diagonal lines describe perfect prediction. The dashed lines indicate a 30 second deviation. **b** Predicted CCS values are presented in comparison to observed CCS values (ground truth). The dashed lines indicate a 10% deviation from the ground truth observations.

small dataset of 409 compounds with RT information and 1346 compounds with CCS values from the MS-DIAL library. It is important to consider the limited size of the training set when interpreting the results. Therefore, the performance was estimated conceptually and not benchmarked against state-of-the-art algorithms. It is worth noting that both RT and CCS values warrant dedicated studies for optimization and evaluation, as demonstrated in the study by Domingo-Almenara et al. on the Metlin small molecule retention time (SMRT) dataset[41].

Figure 5 presents parity plots for RT and CCS values, comparing FIORA's predictions to the experimental measurements. In terms of RT prediction, the majority of RT predictions fall within a 30-second deviation, although a non-negligible number do not. This indicates that the performance is somewhat inconsistent. The $R^2$ value is 0.70. In comparison, a linear model that predicts RT from log$P$ values only achieves an $R^2$ value of 0.63. Log$P$ values for the training and test compounds were calculated using the XLOGP3 program[42]. The results suggest that RT prediction with FIORA's graph model is beneficial over simpler methods, but requires extensive retraining with a larger, homogeneous dataset to be of practical use.

In a similar fashion, predicted CCS values are shown for all three test sets in Fig. 5b. CCS values for CASMI 16 and CASMI 22 compounds could be partially annotated using the MS-DIAL library as reference, although only 2 compounds were found for CASMI 22. The vast majority of predictions fall within a 10% error range. As a means of validation, we compare FIORA to a linear regression model based on precursor m/z, which is a logical proxy of CCS. FIORA consistently outperforms the linear model, albeit by a small margin. The model achieves $R^2$ values of 0.93 compared to 0.9 by linear regression on the test split, and 0.86 compared to 0.80 for CASMI 16 compounds.

Overall, our findings demonstrate that FIORA's modular architecture allows for the seamless prediction of additional compound properties relevant in the context of MS. It remains to be seen whether these orthogonal MS features can translate to improved compound identification rates.

## Significant speed improvements through GPU acceleration
Run time is a particular weakness of fragmentation algorithms, primarily due to the combinatorial nature of potential bond breaks. At the same time, fast processing time is critical to cover the vast chemical space of known structures with spectral predictions and for high-speed (re)scoring pipelines of putative candidates. Table 2 shows the total and average prediction time measured for the algorithms across all test sets. The test environment comprises an NVIDIA A100 GPU, Intel Xeon Gold 6342 CPUs (2.80GHz) and 756 GB of RAM. FIORA runs ~30 times faster on the GPU than on the CPU, and predicts around 10,000 spectra within just five minutes. In terms of CPU performance, FIORA is four times faster than CFM-ID but slightly slower than ICE-BERG. As such, run time improvements are attributed solely to GPU utilization, but are nonetheless significant.

Note that FIORA could be further optimized for run time, e.g., by adding mini-batches to the prediction process. This was realized for the training loop, where 200 epochs of training and validation (without early stopping) were completed in 3 hours. Additional training of RT and CCS values took under 10 minutes. In comparison, ICEBERG was trained on only the positive spectra on a GPU for over 6 days. All in all, by taking advantage of GPUs, FIORA outperforms the other methods by a wide margin in terms of training and prediction speed.

## The impact of single-step fragmentation
Despite the performance benefits of FIORA, its apparent biggest limitation is the shift towards single fragmentation. This section analyzes the impact of this decision on prediction quality. It should be noted that FIORA indirectly covers multiple bond breaks from the same residue, as is explained in the "Fragmentation algorithm" section. Still,

**Table 2 | Run time comparison over all test sets**

|         | Device | Total time  | Spectra predicted | Average time per prediction |
|---------|--------|-------------|-------------------|-----------------------------|
| FIORA   | GPU    | 7m 11s      | 14,100            | 0.03s                       |
| FIORA   | CPU    | 3h 36m 18s  | 14,100            | 0.92s                       |
| CFM-ID  | CPU    | 8h 32m 43s  | 8568              | 3.43s                       |
| ICEBERG | CPU    | 1h 35m 00s  | 8969              | 0.64s                       |

The number of predictions vary due to the specifications of each algorithm. FIORA is the only software that predicted all compounds at all collision energies.

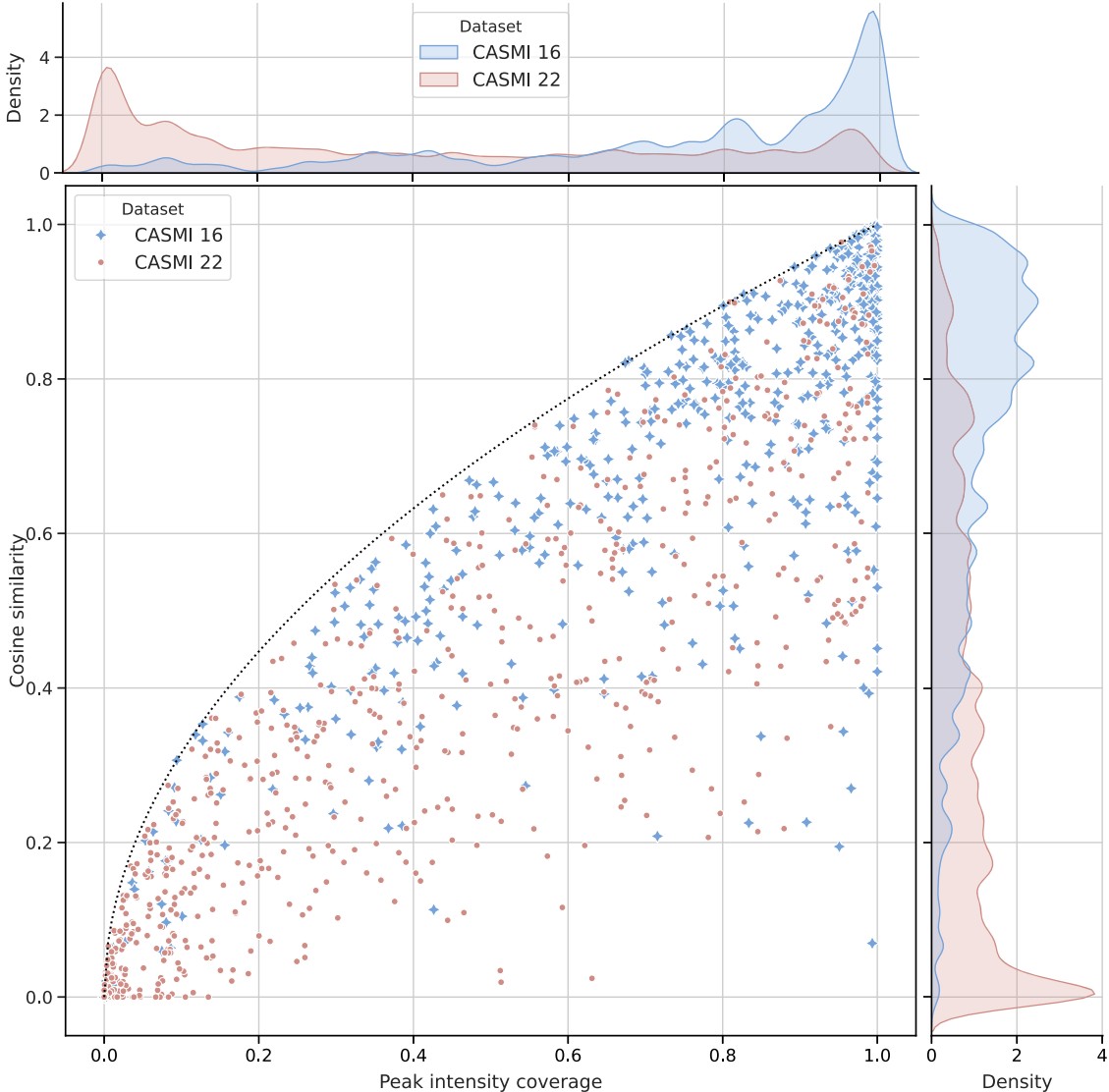

**Fig. 6 | Cosine similarity over peak intensity coverage for predicted spectra from the CASMI 16 and 22 challenges.** Each data point represents the cosine similarity and coverage for a single spectral prediction. The dotted line describes an optimistic upper bound, i.e., maximum cosine similarity at that specific intensity coverage. CASMI 16 represents a standard dataset with high coverage distribution, resulting in overwhelmingly high cosine scores. CASMI 22 represents a rare low-coverage scenario, which leads to low cosine scores.

the model does not account for fragments that break off from different sides of the molecules and does not cleave ring structures. As a result, FIORA does not cover a fragment space as comprehensive as that of CFM-ID or ICEBERG, which may ultimately lead to a significant amount of missed signal for some compounds.

In order to quantify this, we measure the peak intensity covered by FIORA's single-step fragmentation (as defined in Equation (5)) and correlate it to the prediction performance (cosine similarity; as defined in Equation (6)). This comparison is depicted in Fig. 6 for all

predicted spectra from the two CASMI datasets. We observe that the maximum cosine similarity that can be reached by any prediction is bounded by the square root of the peak intensity coverage, as illustrated by the dotted line. This formula can be derived by recognizing that the cosine similarity is maximized if only a single experimental peak (representing the full coverage) is predicted and then computing the resulting cosine similarity. A mathematical explanation is provided in the Supplementary Information. As such, it represents an optimistic upper limit, whereby the maximum cosine

score is lower when peak intensity is distributed across multiple peaks in a more uniform manner. That said, many spectral predictions lie close to the theoretical maximum, indicating that peak intensity prediction is precise, even for cases where the absolute cosine similarity is quite low. In other words, a loss in predictive performance can often be attributed to 'missing signal' rather than poor intensity prediction. Consequently, FIORA's performance is largely contingent upon the level of coverage. In the case of CASMI 16, which has a peak annotation rate of 50% and a median intensity coverage exceeding 80% (see distribution in the top panel of Fig. 6), FIORA is able to simulate many high-quality spectra (compare with distribution on the side panel). For instance, 62% of spectra have a cosine similarity greater than 0.7. In contrast, for a dataset like CASMI 22, where only 25% of peaks could be annotated (median intensity coverage of 32%), cosine scores are correspondingly low. We ascribe the poor performance of FIORA on this dataset to a significant number of spectra with a coverage close to zero. The issue is further compounded by higher collision energies, as documented in the Supplementary Information. Remember that we have already pointed out inconsistencies in the CASMI 22 data in the "Spectral prediction quality" section (and in great detail in the Supplementary Information), so the low coverage is likely amplified by an abundance of noise peaks or poor spectral quality. This hypothesis is supported by the low cosine scores of CFM-ID and ICEBERG, despite multi-step fragmentation. High coverage, as seen in all the other test sets, is directly correlated with a significantly higher prediction quality for FIORA.

While the current implementation is undoubtedly constrained by the limited set of fragment ions, FIORA is capable of effectively compensating through highly accurate intensity predictions. In a way, this reinforces the proficiency of the underlying graph model, which leverages local molecular neighborhoods of the breaking points, in conjunction with a continuous scale of collision energies. Despite single-step fragmentation, FIORA outperforms state-of-the-art methods (refer to "Spectral prediction quality") and we have shown that our approach does not lead to major performance differences between compound superclasses or for structurally distinct compounds. At the same time, it should be noted that single-step fragmentation will restrain FIORA for certain compounds, and it is an important milestone for future improvements.

## Discussion

Experimental spectral libraries are always going to be incomplete in some way[15]. In silico generated spectra can complement these libraries. The nature of fragmentation algorithms makes them inherently valuable for metabolomics beyond spectral prediction alone. Unlike spectral predictors that function as "black box" neural networks, fragmentation algorithms can utilize our understanding of the underlying physical processes and potentially expand our knowledge by learning from experimental data. They can aid compound identification algorithms by anticipating the fragment ion distribution and providing an orthogonal reference to evaluate compound candidates. Bond breaks and fragment ions can be judged individually, allowing for manual and systematic validation of compound annotation. By mirroring the physical fragmentation process, the algorithms lend credibility to the simulated mass spectra, which is critically needed when exploring compounds that are not included in reference libraries.

In this work, we introduce FIORA, an innovative fragmentation algorithm that advances the field in a number of key ways. We show that fragment intensity predictions are significantly improved by modeling bond dissociations based on their local molecular neighborhood. Despite having a smaller set of fragments compared to the state-of-the-art algorithms, FIORA yields higher cosine similarity scores across the majority of test datasets. Notably, already a small

number of graph convolutions, describing short-range structural relationships, produce models with high predictive power (refer to Fig. 2). This achievement should be taken into consideration for all future implementations of in silico fragment ion predictions. We would like to emphasize that our approach of learning local substructures surrounding the bonds rivals global molecular embedding strategies, which are commonly used in recent spectral prediction models, including ICEBERG. Additionally, FIORA incorporates covariates at the fragment intensity prediction level, including ionization mode, instrument type, molecular weight, and collision energy. Collision energy, in particular, has a significant impact on peak intensities as an increase causes the precursor to diminish and new, smaller fragment ions to emerge in the mass spectrum. For example, when modeling the CASMI 2016 data, merging the spectra predicted at the three collision energy steps used in the experiment results in a more accurate simulation than simply predicting the average collision energy (as shown in the Supplementary Information). In fact, the ability to predict a spectrum at any given collision energy is already a notable advancement over the fixed energy levels provided by CFM-ID. Figure 7 depicts the MS/MS spectrum of *Retigabine* accompanied by the spectral predictions of all methods investigated here. The predictions correspond closely to each method's average performance and serve to illustrate the differences resulting from the various implementations. For example, FIORA accurately predicts the precursor intensity and reproduces the most significant peaks. In comparison to the other methods, ICEBERG covers a more extensive range of fragment ions, including numerous low-intensity peaks. However, this broader coverage comes at the cost of reduced accuracy. Conversely, CFM-ID is more specific in its predictions, focusing on a limited number of high-intensity peaks.

Combining the training of positive and negative spectra into a single model allows the algorithm to learn from the fragmentation patterns of the other ionization type. As a result, the performance gain over CFM-ID is even greater for negative spectra (refer to Table 1), despite having twice as many positive spectra to learn from as negative ones. In this study, only [M+H]+ and [M-H]- spectra were evaluated, as they represent the most common types of ionization. However, to demonstrate model adaptability, we have implemented an extension to [M]+ and [M]- precursors in the FIORA-OS model. While there is a slight reduction in prediction accuracy, this is likely due to the very small training and test datasets. Corresponding results are presented in the Supplementary Information. Overall, FIORA was trained on a relatively small dataset of ~10,000 unique compounds, greatly limited by the lack of experimental metadata, especially with respect to the collision energies applied. This might be expanded in the future with better standardization of workflows and ever-growing spectral libraries. Despite the more focused dataset, FIORA generalizes well across different compound classes and to structurally distinct compounds (illustrated by Fig. 3 and Fig. 4). We show that FIORA is capable of learning meaningful representations and compound defining properties with its molecular graph embedding, which is generated as a by-product of the fragmentation process. Based on this embedding, molecules can be clustered at the compound class and superclass level. Simultaneously, the molecular embedding serves as an excellent starting point for the prediction of other MS-related attributes. We conceptually prove this through the prediction of RT and CCS values (shown in Fig. 5). While this feature is still in its prototype stage, it should be considered for the future of spectral prediction software, as it provides valuable dimensions that can help distinguish candidate compounds. Whether it will actually improve compound identification workflows remains to be shown in follow-up studies.

Importantly, FIORA's algorithm maintains a remarkable level of interpretability at the molecular level, while the individual fragment ion predictions are comprehensible in their own right, as they are governed by only a small number of surrounding atoms. This level of

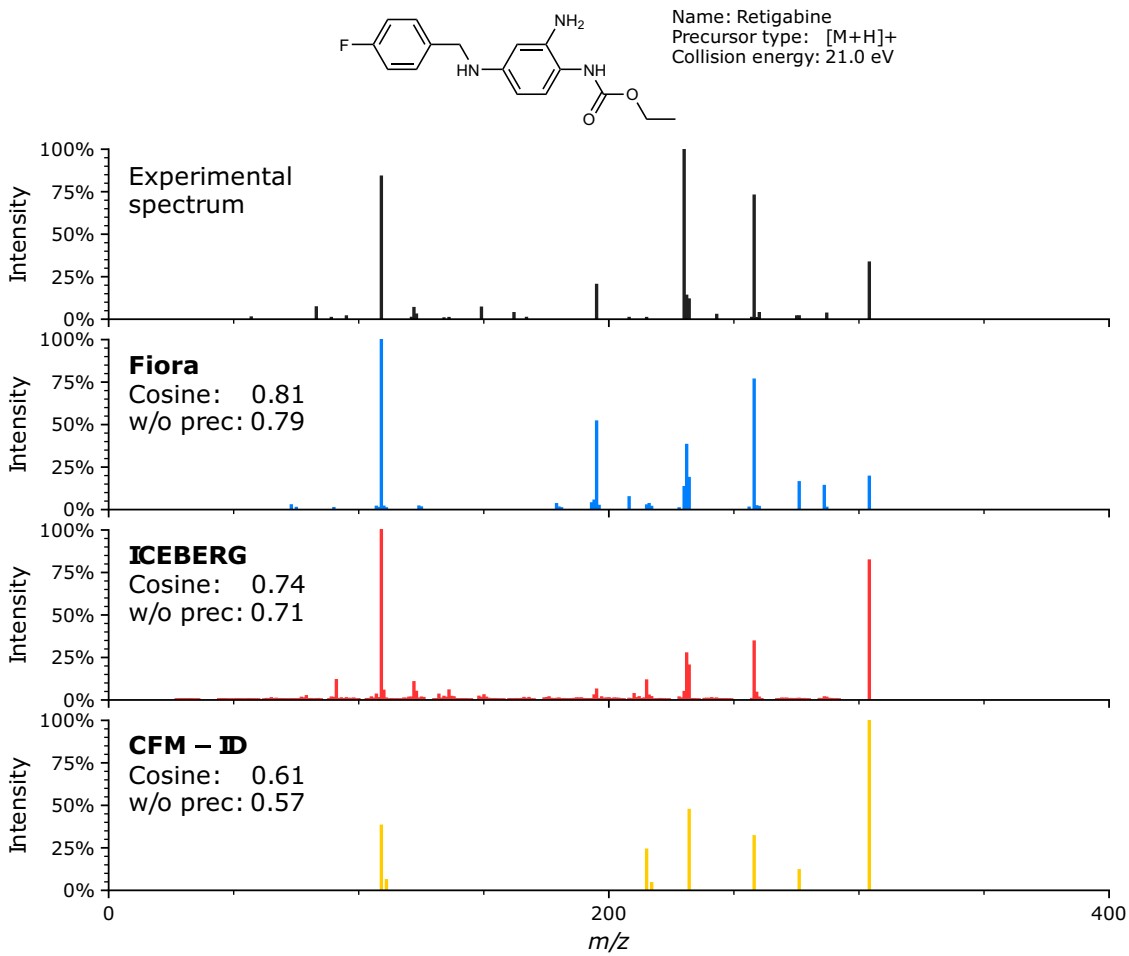

**Fig. 7 | MS/MS spectrum of *Retigabine* showcasing typical prediction performance.** On top, the experimental ground-truth spectrum is displayed; below predicted spectra of FIORA (blue), ICEBERG (red) and CFM-ID (yellow) are presented. The spectrum was selected to reflect the average performance of each method on the test split, with a difference of less than 0.05 in cosine similarity (with and without precursor) from the respective medians. FIORA demonstrates superior performance in estimating precursor intensity compared to its competitors, and accurately predicts all significant peaks, reproducing peak intensities with precision.

explainability is a quality that is often sacrificed in the era of deep learning. In terms of run time, FIORA outperforms CFM-ID and ICE-BERG considerably by taking advantage of GPU-accelerated computations (refer to Table 2) and can be further optimized in the future. Prediction speed is crucial due to the vast space of chemical species, which is poorly covered by spectral libraries.

With that said, no single algorithm is objectively superior to the others in every aspect. On the contrary, this study highlights the advantages and differences between the approaches and closely examines the input, methods, and various aspects of prediction quality. ICEBERG is a fast algorithm that is particularly effective in predicting fragment ions of an average spectrum, i.e., a spectrum merged from multiple collision energies. CFM-ID is based on an 11-year-old algorithm, but remains relevant through a solid statistical foundation and consistent updates. Lastly, FIORA presents a fresh take on bond dissociation, but is limited by single-step fragmentation. In this way, FIORA's fragmentation algorithm could be seen as less effective, since FIORA covers a smaller set of fragments. Indeed, the lack of multi-step fragmentation is currently the biggest limitation. In every other way, FIORA represents the current state of the art in terms of intensity prediction quality and molecular structure modeling. We would like to point out that our implementation leaves open a future extension to multi-fragmentation. The predicted fragments are graphs themselves and can be recursively fed to the model for further fragmentation.

However, it is difficult to link branching fragmentation pathways directly to the observed product ions, i.e., peaks, which serve as ground truth. More sophisticated statistical methods, as for example implemented in CFM-ID, would be required.

In conclusion, advancements in machine learning and the ever-growing spectral libraries are changing the way fragmentation algorithms are trained and deployed. Simulated MS/MS spectra may soon match the quality of experimental libraries and are critically needed to cover the large space of unreferenced chemical species. With this work, we make a pivotal contribution to the field of in silico fragmentation as FIORA taps into the full potential of molecular substructures.

## Methods
### Fragmentation algorithm
FIORA is designed to take advantage of the inherent power of graph neural networks to learn structural patterns and local neighborhoods around chemical bonds. Each molecular structure $M$ is represented as a graph $G$ with atoms for nodes and bonds for edges, which is common practise in computational chemistry. The molecular structure graphs are built from string representations, e.g., SMILES. FIORA operates on neutral molecular structures and only considers information about precursor charge ([M+H]+, [M-H]-) and other covariates at the very end. Fragment ions are modeled by the removal of edges in the graph, indicating singular bond cleavages.

A key concept of our method is that we explicitly model ion rearrangements through hydrogen losses. This is important for direct assignment of peaks to fragment ions and allows end-to-end prediction from the molecular structure graph $G$ to the fragment ion space $\mathbb{F}(G)$. The latter is constructed as follows: Let $E(G)$ denote the set of edges in $G$ and $G_{-e}$ the pair of subgraphs (fragments) that arise from removing edge $e \in E(G)$. The fragment ion space is the set of all subgraphs and fragment ionizations, accounting for up to 4 hydrogen losses, i.e.,

$$\mathbb{F}(G) = \bigcup_{e \in E(G)} \bigcup_{F \in G_{-e}} \left\{ [F+H]^+, [F]^+, [F-H]^+, [F-2H]^+, [F-3H]^+ \right\}.$$

$$(1)$$

For each molecular graph $G$, FIORA predicts the precursor stability $\sigma$ and the abundance values $\theta_f$ for all $f \in \mathbb{F}(G)$. Both are combined using a *softmax* function to compute *fragment probabilities*:

$$p(f) = \frac{\exp \theta_f}{\exp \sigma + \sum_{f' \in \mathbb{F}(G)} \exp \theta_{f'}}. \tag{2}$$

Predicting abundance values $\theta_f$ is conceptually related to the idea of *break tendency* values that were proposed by Allen et al.[22], but here we extend this concept to individual fragment ions $f$. In addition, the precursor stability $\sigma$ allows to model abundances of the intact molecule under various conditions, such as different collision energies. Negative precursor molecules are treated analogously with negative fragment charges.

The MS/MS spectrum is reconstructed afterwards from the exact fragment ion m/z. Figure 8 illustrates the fragment ion prediction process of a single edge break and depicts the information flow in the graph network.

It is important to note that FIORA can be very well used recursively, since fragments are graphs themselves. However, multiple fragmentation steps make it significantly harder to assign ground truth ion probability during training or require other stochastic modeling approaches, such as a Monte Carlo Tree Search. Every assignment of fragment probability (derived from training spectra) bares the risk of introducing conflicts. These arise, when more than one structure can explain a peak, either by having the same or similar weight within the mass tolerance. With multiple fragmentation pathways leading to the same fragment (or fragments of similar weight), the number of conflicts increases dramatically. In favor of speed and more harmonious ion probability assignments we reduce the possible fragmentation events to one. Nevertheless, multi-step fragmentation should be investigated to refine the model in the future. Importantly, multiple bond cleavages from the same residue are already implicitly modeled by directly assigning probabilities to the end-product ions (observed peaks), thereby keeping the missing signal low for the most part. The impact of missing peaks is discussed in the section "The impact of single-step fragmentation".

## Model architecture

FIORA uses a graph network model to predict the precursor stability $\sigma$ and the fragment abundances $\theta_f$ from a molecular graph $G$. Since the fragment space $\mathbb{F}(G)$ is produced from individual edge breaks, edge prediction can be directly used to infer fragment ion abundance $\theta_f$. The prediction of $\sigma$ is based on the whole graph representation.

Initially, atom features are encoded as vectors of integer numbers (for element type, number of hydrogen atoms bound, and ring type information) and are passed to an embedding layer for each number. The embedding dimension of 300 was determined empirically. Bond features are embedded similarly based on bond type and ring type information.

Mathematically, a graph network layer involves a permutation equivariant function $Q$ updating the node (atom) features $X$ into a latent representation $H = Q(X, A)$ using the connectivity of the graph or adjacency matrix $A$. Definitions follow Bronstein et al.[43]. This is achieved by applying a shared permutation invariant function $\phi$ to all $x_i \in X$ and to the features of their local neighborhoods $N_i$, where $\phi$ is a learnable function. In the general case of a message passing neural network, the layer-wise update function can be formulated as

$$h_i = \phi(x_i, \bigoplus_{j \in N_i} \psi(x_i, x_j)), \tag{3}$$

where $h_i \in H$ describes the new hidden representation of node $i$, $\oplus$ is a permutation invariant aggregation operator, e.g., the sum ($\sum$), and $\psi$ is a learnable *message* function. Based on the choice of $\phi$ and $\psi$, various types of graph layers can be modeled.

The type of graph network FIORA uses is customizable. Our implementation is based on the PyTorch Geometric library[44], which provides many graph network architectures. Currently, FIORA supports GCN[45], RGCN[46], GAT[47] and Graph Transformer layers[48]. After each layer an Exponential Linear Unit (ELU) activation function is applied. The choice of graph network and layer depth is discussed in the section "Model selection". We use a 6-layer RGCN as default.

After applying the graph convolutions, the hidden node (atom) representations are taken as input to the final prediction using blocks of fully connected neural networks, which are separate for different prediction targets. For fragment ion prediction, we implement an edge map to concatenate and stack features of all the two node combinations connecting an edge. In addition, the respective edge embedding and covariates (molecular weight, precursor ion mode, collision energy, and instrument type) are concatenated. A fully connected neural network with two layers projects the input onto a 10-dimensional output vector of logits, representing fragment ion abundance $\theta_f$ for the 5 ion modes and for both sides of the modeled edge break. The precursor abundance $\sigma$ is estimated from the entire graph embedding using a global mean pooling aggregation, concatenated with the same covariates. Two additional fully connected layers produce a single logit $\sigma$. All logits (abundance values) are concatenated and passed through a *softmax* function to model precursor and fragment ions as a probability distribution. For model training, a weighted mean squared error (MSE) loss is computed between the predicted and observed fragment probabilities. The latter are estimated in advance by matching peaks to the fragment space using square root intensities. The loss is weighted by 1 over the number of spectra available per compound to reduce the bias towards compounds with multiple entries in the libraries. Network parameters are optimized using ADAM[49].

Finally, the MS/MS spectrum is produced by tracing back predicted subgraphs (fragments) using the edge map and reconstructing the exact peak m/z from fragment weight and hydrogen losses. Ion probabilities are summed to obtain peak intensities because multiple ions can produce the same peak.

RT and CCS values are predicted based on the mean-pooled graph embedding. This is similar to precursor intensity prediction, but with a distinct set of weights and the standard MSE loss function. Hyperparameters were not specifically tuned for this task and training was performed after the fragment ion prediction, freezing all but the relevant dense layers that produce the RT and CCS estimates.

## Data preparation

Experimental MS/MS data of metabolites is diverse. Different data formats and units must be aligned across all measurements, particularly with regards to metadata. Metadata describes experimental conditions and is crucial to understanding and predicting compound fragmentation. We made significant efforts to collect data from various

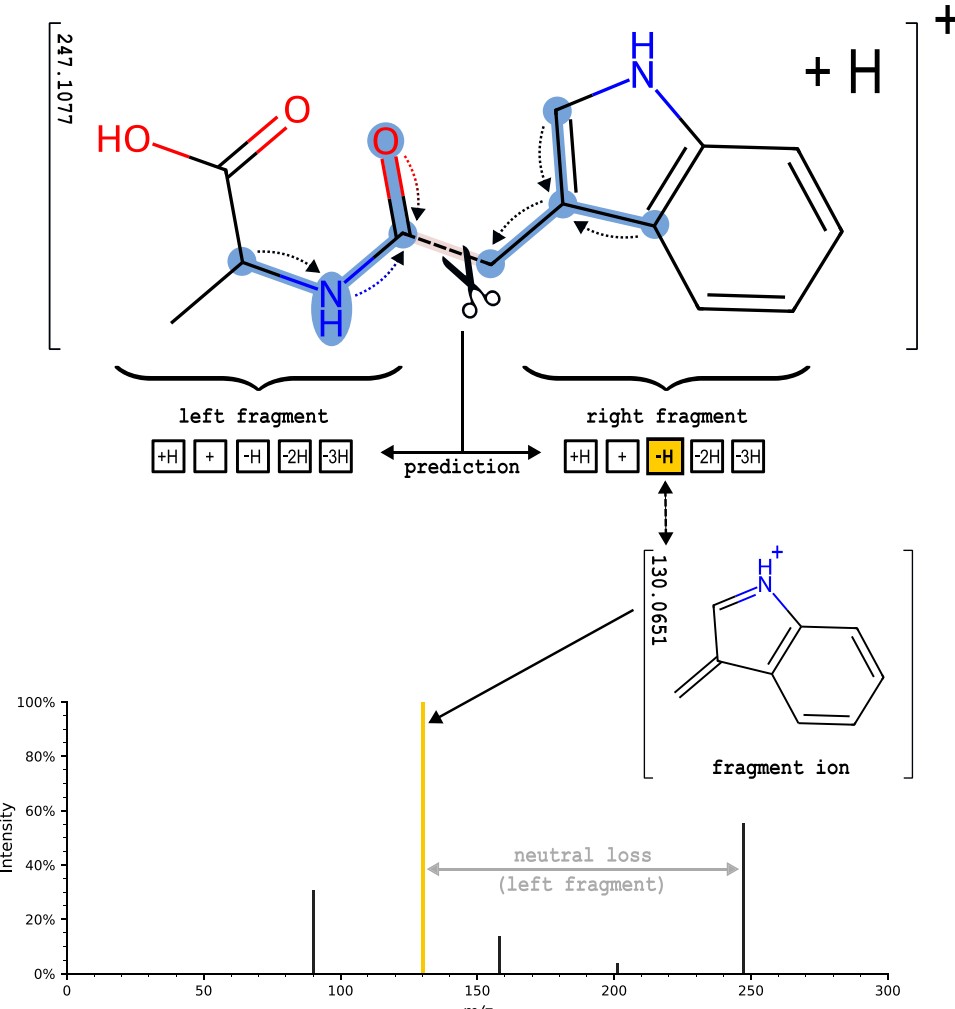

**Fig. 8 | Illustration of FIORA's fragmentation algorithm using the central bond as an example.** FIORA learns the local neighborhood of the bond over several graph convolutions, as illustrated by the dotted arrows indicating the information flow. The learned molecular structure is outlined in blue. For visual clarity, only two graph convolutions are illustrated and arrows are directed towards the designated bond. Based on the bond features and surrounding substructure, fragment ion abundances (and neutral losses) are predicted. In this case, the prediction suggests a loss of two hydrogen atoms and the formation of a new double bond in the right fragment. Peak probabilities are reconstructed statistically considering precursor stability and the abundances of all other fragments.

sources and align the information as accurately as possible. This enables our model to utilize covariate information, such as collision energy or instrument type, but requires extensive data pre-processing. Nevertheless, data preparation is a critical factor influencing model performance. The following section provides detailed information on the most important pre-processing steps.

Our setup relies on three spectral libraries. The first is NIST'17[35], which contains a large collection of MS/MS spectra acquired from authentic compound standards. The library was converted to .msp format using the *lib2nist* program that is included with the NIST library. Compound information was exported in the .MOL format and then parsed back into the spectral library using the Python *rdkit* package. For a detailed walkthrough of all library parsing steps, please refer to our script on GitHub (found at https://github.com/BAMeScience/fiora/blob/main/lib_loader/nist_library_loader.ipynb)[34]. It is important to note that NIST'17 is a commercial library requiring a license, which severely limits its public use. The homogeneity of the data may also limit its ability to generalize to more diverse experimental setups. The NIST'17 MS/MS library does not provide information about RT and CCS values of the measured compounds.

In addition to NIST'17, we process a public library available in .msp format on the MS-DIAL website[36], which contains a collection of

annotated MS/MS spectra from various other spectral libraries, such as MassBank and GNPS. Unlike most of the public libraries, the MS-DIAL spectral library standardizes metadata, making crucial information about collision energy, RT and CCS readily accessible.

Lastly, we use the recently published MSnLib v1.0 (https://doi.org/10.5281/zenodo.11163381)[37], which contains over a million multi-stage fragmentation spectra, including 177,390 MS/MS spectra for 15,967 unique compounds. An overview of all three spectral libraries can be found in Supplementary Table 5.

All libraries are pre-processed in a similar fashion, converting normalized collision energies (NCE) to electron Volts (eV) where possible, using the formula from the Proteomicsnews blogpost (https://proteomicsnews.blogspot.com/2014/06/normalized-collision-energy-calculation.html)[50]. MS/MS spectra with missing collision energies or unclear formats were excluded. Compounds with inconsistencies between SMILES, MOL format and InChiKeys were filtered out immediately. In addition, we filtered for ionization types [M+H]+ and [M-H]- (in case of MSnLib [M]+ and [M]- ions were considered as well) and imposed requirements for spectra to be selected for training. These requirements included a maximum weight of 1000 Da and maximum collision energy of 100 eV. Compounds were fragmented by single edge removal, and peaks were matched at 50 ppm (10 ppm for

MSnLib) to the set of fragment ions described in the "Fragmentation algorithm" section. Note that the fragment tolerance for NIST and MS-DIAL is quite lenient, since high machine accuracy could not be guaranteed for all spectra. For ambiguous fragment matches that can explain the same peak, equal probability was assigned to each fragment. The fragment matches are then used to determine the final data set containing spectra suitable for training. We enforced at least two peak matches, a minimum peak intensity coverage of 50%, and a maximum precursor intensity of 90%. Then, soft filters required that each spectrum have either 50% of the total peaks or at least five peaks matched by fragments, or an exceptionally high intensity coverage of 0.8. This ensured that there is sufficient overlap between the theoretical fragmentation patterns and the observed peaks. We combine the NIST and MS-DIAL libraries into a single training library for the standard FIORA model. In this way, the library has a size comparable to that of CFM-ID's training set, comprising 74,401 spectra from 10,692 compounds. Additionally, we prepared the MSnLib data to train a separate fully open-source model.

### Training and testing

FIORA was trained on 80% of the molecular structures, with the remaining 20% split evenly between validation and testing. The training process consisted of 200 epochs using the ADAM optimizer, a weighted MSE loss and a scheduler that reduces the learning rate upon reaching a plateau in validation loss. The model checkpoint with the lowest validation loss was loaded afterwards. Hyperparameters were tuned mostly empirically using the validation set. The types of graph layer and model depth are systematically evaluated in the "Model selection" section. Exact model specification can be found in the Supplementary Information.

ICEBERG was trained using identical training and validation splits, and following the training steps provided at the original GitHub repository (https://github.com/samgoldman97/ms-pred). We used the commit from October 21, 2023, which is slightly newer than the 1.0.0 release, as it contains a detailed retraining workflow. In the case of CFM-ID, we used the pre-trained model v4.4.7 running with a docker container provided on Docker Hub (https://hub.docker.com/r/wishartlab/cfmid). Retraining CFM-ID with our dataset was found computationally infeasible. An open-source version of FIORA was trained on the MSnLib spectra with the exact same model specifications.

In addition to the 10% test split from the NIST'17/MS-DIAL library and the 10% test split from MSnLib, we selected the CASMI 2016 and CASMI 2022 datasets for benchmarking. Both datasets were downloaded from the CASMI contest webpage at http://casmi-contest.org. We selected [M+H]+ and [M-H]- precursors and set covariates according to the descriptions. In each case, as many compounds and spectra as possible were extracted, which includes the priority and bonus challenges for CASMI 22 and training as well as challenge spectra for CASMI 16. In the CASMI 16 challenge, a stepped collision energy was used, so we predicted spectra with FIORA for all 3 collision energies and merged them into a single "stepped" spectrum. CFM-ID predicts compound spectra at fixed collision energies of 10 eV, 20 eV and 40 eV. Therefore, we evaluated the prediction with the closest matching collision energy, following the approach of Wang et al.[23]. For CASMI 22, the extraction of the challenge spectra was more complicated. The OpenMS[51] library using its Python wrapper was used for reading the *mzml* file and for the extraction of the collision energy levels. Challenge spectra were extracted using a precursor tolerance of 10 ppm and retention time window of 5 seconds. Multiple measurements at the same collison energy were merged into a consensus spectrum. Despite our efforts, not all collision energies for all challenge spectra were found. Reducing the stringency of the tolerance values did not solve this, as it lead to ambiguous matches to more than one compound. All processing steps are found on our GitHub (https://

github.com/BAMeScience/fiora/tree/main/lib_loader)[34]. To preserve the integrity of the test results, all compounds from the test sets were removed from the training data for FIORA and ICEBERG. CFM-ID was trained on the METLIN[17] library, which means that this separation is not guaranteed.

### Evaluation metrics

Cosine similarity (Eq. (4)) measures the similarity of two vectors, $A = (a_i)$ and $B = (b_i)$, in the inner product space. It is defined as the cosine of the angle $\rho$ between the two vectors:

$$\text{Cosine similarity}(A, B) = \cos(\rho) = \frac{A \cdot B}{\|A\| \|B\|}. \tag{4}$$

Furthermore, we define the coverage between $A$ and $B$ as:

$$\text{Coverage}(A, B) = \sum_i a_i \mathbb{1}_{b_i > 0} \tag{5}$$

To obtain the cosine similarity and coverage between MS/MS spectra, they need to be discretized along the m/z dimension. This is typically done by binning m/z values or by matching query and reference peaks and then assigning a dimension to each peak pair and each unmatched peak. The latter is specified in Equation (6), which defines the cosine similarity between the spectra $S_A$ and $S_B$ at a tolerance value $t$ that determines whether two peaks match:

$$\text{Spectral cosine similarity}(S_A, S_B, t) = \frac{\sum_{(mz_k, I_k) \in S_A} \sum_{(mz_l, I_l) \in S_B} I_k I_l \mathbb{1}_{|mz_k - mz_l| \leq t}}{\sqrt{\sum_{(mz_k, I_k) \in S_A} I_k^2} \sqrt{\sum_{(mz_l, I_l) \in S_B} I_l^2}}. \tag{6}$$

Peaks are represented as tuples of m/z and intensity values $(mz_k, I_k)$. Note that Equation (6) reflects the cosine angle only if peak matches are unique. This is not necessarily the case as multiple fragment predictions may have the same or very similar m/z. Therefore, intensities are summed within the tolerance window defined by $t$, only for the purpose of spectral scoring. The spectral coverage of peak intensities is defined analogously based on the tolerance $t$.

Although cosine similarity expertly describes the angle $\rho$ between the spectral vectors in an inner product space, it has some practically shortcomings. Dominant (high-intensity) peaks have a dramatic effect on the cosine similarity, regardless of whether other low-intensity peaks are matched. As a result, the vanilla cosine similarity may not be indicative of how well low-intensity fragmentation patterns are reflected. This problem is exacerbated by a small number of peaks. Solutions are hyperscores, which take into account the number of matched peaks and are used in proteomics[52] or logarithmic or square root transformations of the initial peak intensities, which even out differences in peak intensity and de-emphasize dominant peaks. In this study, all results use the cosine similarity of square root transformed peaks, as is common practice in metabolomics[53]. The evaluation was conducted in two parts: cosine similarity over the full spectrum and cosine similarity without the precursor peak. The former is a more accurate reflection of the quality of the models as a whole, taking into account the entirety of the prediction space. However, the precursor peak is shared by all molecules with the same formula, and hence is not specific to the underlying molecular structure. Consequently, it carries less information relevant for metabolite identification and is typically excluded during the MS/MS matching step of spectral library search. Thus, cosine similarity without the precursor peak offers a more faithful representation of the models' practical applicability. It is important to note, however, that precursor intensities still encapsulate valuable information about molecular stability. Furthermore, the consequences of precursor removal must be carefully evaluated. For

instance, the influence of noise and measurement inaccuracies on the resulting cosine scores may become more pronounced following the removal of a dominant precursor. Deviations in peak intensity that are minimal in relation to the total ion count may suddenly result in considerable relative differences between the experimental spectrum and the prediction.

It is crucial to recognize that any modification to cosine similarity is arbitrary and that alternative similarity metrics may prove equally effective or superior for the purpose of compound identification. In a previous study, we showed that the standard cosine similarity (with square root intensities) for predicted spectra is deceptive in some cases and leads to high scores for false spectrum matches[54], albeit for peptide spectra. In metabolomics, many of the same principles still apply when assessing spectral similarity for small compounds. A metric such as the cosine bias, described for spectra by Lam et al.[55], is a good indicator of how dependent the cosine similarity is on matching a few dominant peaks. Ideally, a high quality prediction would have a high cosine similarity and a low bias due to many matching peaks. We have suggested a bias-adjusted cosine score before. However, many compounds spectra have only a few peaks and even a perfect match might result in a high cosine bias, making it challenging to fine-tune scoring functions. We have monitored the bias for different versions of the cosine similarity in the Supplementary Information.

In summary, as datasets and search spaces grow rapidly, refined similarity scores will be necessary to distinguish true spectral matches from false ones. This is especially true, since prediction software might not cover the complete fragment space of experimental spectra. At this point, we evaluate prediction quality solely on the standard cosine similarity using square root transformed intensities with and without the precursor peak.

### Reporting summary

Further information on research design is available in the Nature Portfolio Reporting Summary linked to this article.

## Data availability

The computational data generated in this study have been deposited in the Zenodo database [https://doi.org/10.5281/zenodo.14782250][56]. Unless otherwise stated, all data supporting the results of this study can be found in the article, supplementary, and source data files. The data from public spectral libraries used for training and testing are hosted on the MS-DIAL website [https://systemsomicslab.github.io/compms/msdial/main.html][36] and, in the case of MSnLib, are archived on Zenodo [https://doi.org/10.5281/zenodo.11163381][37]. Training, validation, and test splits are additionally available on Zenodo [https://doi.org/10.5281/zenodo.14782250][56]. In the case of the NIST'17 library, access is restricted due to its commercial licensing[35]. Source data are provided with this paper.

## Code availability

The code supporting this study is available on GitHub [https://github.com/BAMeScience/fiora] and archived on Zenodo [https://doi.org/10.5281/zenodo.14651773][34]. This repository includes all relevant scripts, open-source model weights, and Jupyter notebooks used for training, testing, and producing the figures in the manuscript.

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

## Acknowledgements

The authors thank Justin van der Hooft, Niek de Jonge, Tanja Holstein, and Sasan Amariamir for all their helpful and stimulating discussions. Additionally, they sincerely acknowledge Ferdous Nasri for proofreading the manuscript.

## Author contributions

Y.N., P.B., and T.M. conceived the initial idea for the method. A.K. assisted with the mathematical conceptualization. The analytical methods and results were verified by F.R and J.L with particular focus on the chemical and metabolomic aspects. K.R. aided the project with his expertise in algorithms. Y.N. implemented the algorithm, wrote the manuscript, and created the visualizations. All authors discussed the results and contributed to the final manuscript.

## Funding

## Competing interests

The authors declare no competing interests.
