## [Transparent Peer Review file · Nature Communications]

Fiora: Local neighborhood-based prediction of compound mass spectra from single fragmentation events

Corresponding Author: Dr Philipp Benner

Version 0:

Reviewer comments:

Reviewer #1

(Remarks to the Author)

Strengths

- * This paper presents a neural network to perform single-step bond-breaking for in silico mass spec fragmentation. The effort of improving the efficiency of existing models is a nice effort to make in silico fragmentation models more accessible for the community.
- * Considering bond-breaking and predicting two fragments jointly is interesting and sounds novel.
- * The experiment study is extensive and detailed.

Weaknesses

- * One of the shortcomings of the experiment part is that all models are trained on an older version of the NIST library (NIST'17), given the fact that the baseline method ICEBERG was trained on the newer NIST'20. Are there any justifications for why NIST'17 is considered? It will be a more fair comparison with the baselines if the experiment is performed on NIST'20 (also where ICEBERG is optimized for, in case of any potential overfitting issue on the smaller NIST'17). Also, method developers should always work with the best dataset available to make sure that the downstream user is getting what they expect.
- * The authors claimed that the support of both $[M+H]^+$ and $[M-H]^-$ is a novelty. But it is also worth noting that this paper ignored some other common adducts such as $[M+H-H_2O]^+$, $[M]^+$, $[M+Na]^+$, which are considered in other baselines. Is there a specific reason why they are not covered? It seems a natural extension to just add more dimensions to the one-hot encoding of adducts.
- * The use of single-step bond-breaking, as also discussed by the authors, is probably the major technical contribution and potentially a major bottleneck in performance. It would be better to discuss:
 - What is the exact proportion of peaks that cannot be explained by single-step fragmentation?
 - How does the model break rings?
 - The authors mentioned that "Fiora implicitly covers multiple bond breaks", is there an example of that?
- * The introduction of retention time (RT) and collision cross section (CCS) is interesting, but the reported results in this current version seem just proof-of-concept and the accuracy and scalability at the current stage seem not ready to be deployed in real-world tasks.
- * The clustering effect shown in the embedding of Figure 4(a) is a bit trivial, as has been found in many GNN-based studies of molecules. It seems more like the expressive power of GNN itself, instead of any new improvements brought by learning with the mass spec. It might be interesting to see if there is any correlation between GNN embedding similarities and mass spec similarities.
- * It may need more care when comparing the training speed. The authors mentioned that "ICEBERG was trained on only the positive spectra on a GPU for over 6 days", while it is worth noting that ICEBERG's code involves many evaluation steps that are not correlated to training. They also run replicate experiments on 4 splits. It needs to be double-checked whether ICEBERG's single "training" step is taking as long as 6 days. Also, it is a good idea to report the CPU/GPU/RAM/etc of the testbed for readers to understand the computational resources needed to run the model.

(Remarks on code availability)

Reviewer #2

(Remarks to the Author)

The authors present a novel approach for the prediction of high-resolution tandem mass spectra, based on a graph neural network approach. The approach is based on predicting intensities for all single fragmentation events arising from a single fragmentation event. The paper is well written and to the point. The code is freely available. The authors openly discuss strengths and are honest about the limitations (single fragmentation) of their approach. Collision energy as a continuous input value is a nice addition over many existing models. In summary, nice work and certainly worth publication.

My only serious criticism is about the choice of not excluding precursor peaks from spectra for evaluation. The authors argue that:

"Other adjustments to the cosine score include removing the precursor [... This] may eliminate a dominant precursor peak in some cases, but significantly increases the overall cosine bias. Second-order dominant peaks may arise from obvious losses, such as H₂O, and the molecular stability and the relationship of the intact molecule to the fragments are a indispensable parts of the fragmentation pattern. This makes precursor removal rather futile for spectra that already contain very few peaks [...]"

I strongly argue that the authors should add a second evaluation with removed precursor. The unfragmented precursor mass is not structure-specific for a candidate of the same formula, and it is common practice to exclude precursor from spectral library search for that reason. (A small neutral loss, such as H₂O, at least indicates something about the structure). For this, it doesn't matter if the spectrum only has few peaks. It is obvious that not excluding precursor inflates the cosine score for low collision energies (as is easily seen in Supplementary Fig 3 and 4). We care less about cosine bias than about seeing whether the model can solve the hard task, namely predicting the fragmentations. If a model can show surface-level strong performance numbers by predicting unfragmented precursor intensity (which would be excluded in search anyway), this obfuscates the real performance of the model for the cases we really care about and that really make a difference. I don't care about the cosine being nominal 0.8 or 0.7, as this is such a dataset-dependent number, I care about the performance on a *hard* problem compared to baselines and SOTA.

(As we know, model evaluations should be such that they are "hard but not too hard" for a model. Böcker 2017: "In summary, a challenge must not be too hard, or all methods will fail. But if you evaluate a method in its 'comfort zone', there is not much to be learned. If you want to see how strong a piece of concrete is, you have to break it.")

Other than this, I have only few comments:

* General: Particularly driven by the interpretation Fig 4b, it would be good if (all) box-whisker plots had the number of class instances (n=??) for every bar.

* P8

"Still, the relative loss of performance loss is lower..."

I wouldn't read too much into this honestly, as the magnitudes are small.

* P9, 10 RT/CCS: As presented, it is a bit of a gimmick feature. It would be more substantial if compared to some baselines, at least very simple ones such as logP for RT and pure m/z predictor for ccs, to show that it's better than just throwing darts or drawing a rough line. Preferably should compare to models closer to the state of the art.

* P11, comment on CASMI22:

Obviously these spectra are somewhat strange. The cynical reading of this is, however, that FIORA is good at low CE because it predicts the unspecific precursor, and bad at high CE because it misses multiple fragmentations. Just another reason to add a validation with precursors removed. It is still a consolation that everyone is bad at CASMI22.

* P17, Table 3: I commend the authors for the diligent filtering. Kind of sad numbers in the end, compared to the nominal size of those libraries. (Makes me wonder whether the MS-Dial includes the up-to-date GNPS and MassBank. Maybe try to get the cleaned-up current GNPS by Huber/deJonge and the current MassBank 2024.06 release. Still, it's clear that NIST will give better results anyway.) I would be interested (maybe in the SI) in a step by step reduction of cpds / spectra after each filtering step.

* P18

"Placing additional emphasis on higher ion masses, as suggested by Stein & Scott (1994) [51], for instance, by multiplying peak intensities by m/z values, makes sense in order to increase the score for more compound-specific fragments. Fragments ions with low masses are more likely to be shared between different compounds."

This adjustment was developed for GC-MS data, where the precursor mass is not known. It cannot be transferred to the ESI-MS/MS case where the largest mass (isolated precursor) is actually the least specific by definition, and other large masses (like neutral losses) are also relatively unspecific, as the authors themselves note above. (Not that it matters much, since the authors didn't include such an adjustment.)

(Remarks on code availability)

I haven't reviewed the code in detail, but gave it a quick glance and made sure I can install and run it. I tested the version from the preprint, which installed fine, but there are no weights included because they rely on closed-source data, so one would have to retrain or ask the authors.

Reviewer #3

(Remarks to the Author)

This is an interesting study. I have always wanted to implement a similar approach, so I enjoyed (in part) learning from the results of this study. It is very commendable that the authors have compared their algorithm with a yet-to-be-published tool reported in a preprint. I appreciate that the authors want to demonstrate the ability of FIORA to be used with RT and CCS, and I also understand that they want to play that part down since it is not the main paper focus. However, the paper is 'obscure', meaning that has parts that are hard to understand, that it has some incongruences, and that some parts are described in a highly technical way, not suitable for the wide audience where articles in Nat Comms are written for. Generally, the write-up could be condensed to the important points. It could also benefit from additional results and metrics. I recommend the authors take a look at the following paper: 10.1021/acs.analchem.2c02093, which uses intuitive metrics to assess and compare the performance of the prediction algorithm, including the top k performance (a metric used in ICEBERG as well).

Some of the 'obscure' things that I observed:

- From what is understood from the main text, the authors report the main algorithm results and comparisons with the cosine similarity. At the very end of the paper, in the methods section, the authors then state that they use the square root transformed peaks, something that the authors claim is common in metabolomics (without a reference).

- The authors mention that they have "retrained" (literally) the ICEBERG algorithm with the same dataset used for FIORA, but then they contradict themselves. It seems that the authors use the metrics reported from the preprint of ICEBERG, but ICEBERG seems to be trained with NIST20 instead of NIST17.

- Fig 2 and Fig 3, Fig 2 says that the vertical lines correspond to the 95% CI. For Fig 3 this is not defined, and I assume that these are the same. If that is correct, what these figures show is very hard to believe, especially from what is then observed in Fig 6. It implies the algorithm has incredible precision. Please report the accurate 95% CI or at least the IQR.

- "Out-of-distribution bias": a complicated way of saying lack of generalization (a term later used by the authors).

- The whole section about speed is not clear: it is faster only on the GPU? Table 2 is misleading, as it seems that a CPU is not faster, and it is not clear if Table 2 results are CPU or GPU.

- The section's impact on single-step fragmentation is also cryptic and hard to follow. E.g., how is the dotted line calculated?

- The manuscript does not have paragraphs or page lines, which hampered the job of this reviewer.

- What is more surprising is to read sentences in the results section like: 'overall stellar performance', 'it is all more impressive', 'it is remarkable'.

Other concerns that I have:

1. Collision energy (CE) is not included or discussed in the results, especially given the singularity of FIORA using the one-atom break strategy, and given also the importance of CE in MS/MS comparison in metabolomics. First, it is not clear at all which CE does the algorithm cover. Second, I would like to see how FIORA fails at predicting low-intensity peaks as the CE increases. Overall, some mirror plots for the prediction, similar to the ones used for CMF-ID, ICEBERG, or in 10.1021/acs.analchem.2c02093, could improve the results section.

2. I appreciate the results from Fig 3, but the authors could use the GNPS library to demonstrate the generalization capability, as it is also quite different from the data contained in NIST/MS-DIAL and is much larger than CASMI and widely used.

3. Fig 6 could be plotted with the results from the validation set, as it makes more sense given the high dissimilarity between the CASMI data and the training data.

4. The results for the comparisons made against ICEBERG and CFM-ID should be shown with more statistical rigor: cosine similarity boxplots for the three algorithms. I understand the rationale behind the use of the root square cosine, but it would be interesting to see the comparison made with plain cosines (unweighted). What about using other metrics like entropy (10.1038/s41592-021-01331-z)?

5. It is not described in the methods section (Fig 7) how ambiguities are solved: e.g., in Fig 7, one experimental peak could correspond to the formula C₄NO₁ (without considering the H). This formula could come from two possible substructures (i.e., there are two different groups of nodes whose atoms could correspond to C₄NO₁). How do the authors know from which of the two possible substructures this peak stems from?

6. Data preparation: the authors state "converting NCE to eV". How did the authors approach this? Why such a large ppm error (50 ppm) was used? "90% of precursor intensity" was used at most? I do not understand it. But if I guess correctly, this would limit the inclusion of spectra acquired at high CE, as higher CE yields smaller precursors.

7. The trained model object does not seem to be published in the Github repository, hampering the wide adoption of this tool by the community.

(Remarks on code availability)

Version 1:

Reviewer comments:

Reviewer #1

(Remarks to the Author)

I really appreciate the authors' response and the thorough update to the manuscript. After reviewing the revised submission and reading the comments from other reviewers, I still have a few remaining questions and concerns:

* I understand that the authors used NIST'17 due to licensing constraints, and I appreciate the clarification. However, I just wanted to note that not using the latest dataset might impact the perceived technical novelty of the work. For anyone who may not be familiar, NIST licenses typically cost around \$2,000.

* In my initial review, I didn't notice that the median is used instead of the mean. Could you clarify the reasoning behind this choice?

* The claim that "Fiora implicitly covers multiple bond breaks from the same residue" is a bit unclear to me. When reading the manuscript, I interpreted this as "Fiora has the ability to break multiple bonds beyond single-bond breaking," but the authors have confirmed this isn't the case.

* In your response to Reviewer #3, I noticed the requested top-k retrieval study wasn't included. Instead, you mentioned that cosine similarity (without the precursor peak) serves as a reasonable proxy for the top-k retrieval recall rate. I respectfully differ in opinion here; the top-k retrieval study is a well-recognized metric in real-world applications of your model and has been utilized in both the ICEBERG and CFM-ID papers. It offers a more direct and convincing assessment than cosine similarity.

Additionally, I want to make a note on the training speed of ICEBERG — not as a "weakness," but just as a point of comparison. I've run their open-source code myself, and while I don't recall the exact training times, I remember both scripts finishing significantly faster than six days. Also, the ICEBERG paper mentions, "Models are trained on a single RTX A5000 NVIDIA GPU (CUDA Version 11.6) in under 3 hours for each module." It may sound like a simple question, but did you double-check that GPU resources were fully utilized for training? I know deep learning model setups can be challenging.

(Remarks on code availability)

Reviewer #2

(Remarks to the Author)

The authors now included the important analyses without precursor peak, thank you! Also great to have open weights. No further comments.

(Remarks on code availability)

Reviewer #3

(Remarks to the Author)

I thank the authors for their thorough revision and for answering my concerns. I believe the manuscript has been greatly improved. I only have a minor comment:

Regarding the CI intervals. Thank you for the answer, indeed, the issue here is that the CI is low due to the large number of data points. I strongly believe that providing the IQR is a more insightful alternative. Although I understand that authors want to show the best possible side of their results, it is fine to have a large variation, and it gives the readers a more transparent view regarding the expected error range of their predictions. At least, include the IQR range as a supplementary figure (and referenced in the manuscript). This applies to Figure 2, but especially 3. Regarding the best way to present the results, I find supplementary figures 8 and 10 the most convincing in showing Fiora's superiority to the other models.

The same applies to SI Figure 16, I would like to see the IQR for this one as well. The CI shows that, comparatively, FIORA is superior, but I am interested in knowing what variation I can expect in my predictions if I use FIORA.

(Remarks on code availability)

Version 2:

Reviewer comments:

Reviewer #1

(Remarks to the Author)

Thank you for your response and the updates. My evaluation of the manuscript at its current stage is as follows:

- 1) The manuscript presents technical novelty that I believe will be of interest to the community.
- 2) However, as a deep learning paper, I think it is important to include performance metrics that have been used by recent peer methods. This aligns with the points I raised in my comments and the authors' response to NIST'20. The proposed MSnLib is great, of course, but there seem no previous methods being evaluated on that.

(Remarks on code availability)

Response Letter

Dear Editor, dear Reviewers,

We sincerely thank the reviewers for their generous feedback and expertise on the subject matter. In this response letter, we provide a detailed account of all changes made to the manuscript and software, addressing both major and minor revisions. We have also appended our point-by-point responses to each comment. With your guidance, we believe the manuscript has been significantly improved and will contribute meaningfully to the field of metabolomics.

A key concern raised by all reviewers was the data, particularly regarding model performance and practical applicability. In response, we have incorporated a novel spectral library, MSnLib (Brungs et al., 2024), into the study. We used this open-source library as an additional test set, comprising over 1,000 compounds that are structurally dissimilar to the training data, thereby nearly doubling the test space. Performance is now tracked across four different datasets in both positive and negative ion modes.

In addition, we trained a second model, designated Fiora-OS, on over 8000 compounds from MSnLib. Fiora-OS is completely open source and readily available on our GitHub repository. This is accompanied by several software updates, significantly improving usability, model performance, and robustness. Further details are provided below. In doing so, we were able to demonstrate Fiora's quality and broad applicability on yet another – even more challenging – dataset, while also providing a free model for immediate use by the community.

Moreover, an additional performance evaluation has been included in the manuscript, focusing exclusively on structure-specific fragment peaks (by omitting the precursor). This approach assesses each model's ability to differentiate between molecules that share the same precursor weight. The sections discussing spectral prediction performance, model generalizability, and its limitations have been substantially revised. In particular, compound distributions and covariates, such as collision energies, are discussed in greater detail.

Nearly all the figures have been updated to reflect the new results and to include additional information. New spectral prediction plots have been added to visually convey the models' outputs and performance in a concise and comprehensible manner. An overview of all the changes, including several minor revisions, is provided below. We would also like to highlight the addition of several new chapters to the Supplementary Material, which now includes 19 figures and 4 additional tables, further reinforcing the quality and robustness of our method.

We believe that the revisions have resolved all major concerns, enhancing the quality of the manuscript. Thank you again for your valuable insights and careful consideration.

On behalf of all authors,

Yannek Nowatzky

Change log

Data Changes

- We retained the original testing scenarios but supplemented them with additional data. Specifically, we processed the novel open-source spectral library, MSnLib, which contains 177,390 MS/MS spectra for 15,967 unique compounds. After applying rigorous filtering and quality control measures, we generated a refined dataset of over 10,000 compounds. This dataset was then split, with 80% allocated for training and 20% for validation and testing. Data analysis details can be found in Table 4 of the manuscript and as part of the fully available script on GitHub. Importantly, the MSnLib test set consists of compounds that are distinct from those in the original training set, as reflected in the low Tanimoto similarity scores illustrated in Fig. 14 in the Supplement.

Model Changes

- A new open-source version of Fiora was trained using MSnLib data and is now the default model upon installation. This allows users to install the package and run predictions in under 5 minutes without requiring further technical expertise. While the open-source model differs from the NIST/MS-Dial model, it maintains comparable accuracy, as detailed in the main manuscript.
- Training is now based on square root intensities, reducing the emphasis of the MSE Loss placed on dominant peaks. Note that spectral prediction output still produces untransformed peak intensities. We found that this change significantly improves the accuracy of reproducing smaller fragmentation patterns and results in a more diverse set of peaks. This increases model robustness, reducing the reliance of cosine scores on singular dominant peaks, e.g., the precursor.
- Predictions for both [M]⁺ and [M]⁻ spectra are now also supported by the new model.
- Updates have been made to exception handling, usability, and unit tests, further enhancing the software's reliability.

Manuscript Changes

- A new stacked spectral prediction plot, similar to mirror plots, has been added to both the manuscript and supplement. This figure allows for direct comparison of prediction examples across different tools (cf. Figure 7).
- In addition to the original spectral quality analysis, prediction performance is now evaluated with the precursor removed. In this way, cosine similarity focuses solely on fragment peaks to better reflect practical model performance. This expanded discussion continues in the Supplement, with further scores described and figures showing side-by-side comparisons of cosine similarity distributions across the tools.
- A more robust evaluation of RT and CCS predictions has been conducted. RT predictions are now compared to linear regression (LR) based on logP values, serving as a baseline. CCS are compared to the LR estimate based on precursor m/z, as before. The section has been overhauled and made more concise to emphasize the "Proof of Concept" status of this content.
- Collision energies are now discussed more thoroughly, regarding model performance and with respect to precursor intensity. This is complemented by a long discussion in the Supplementary Material, showing prediction examples at various collision energies and rigorous statistical analyses.
- Unique compound counts have been added to box-whisker plots, presented via histograms in a panel above (cf. Figures 3 and 4).
- Confidence intervals and standard deviation is more clearly explained (cf. Fig. 2 and 3).

- The single-step fragmentation section has been completely overhauled, with the proportion of annotated peaks now being tracked. Explanations are concise and clear, and specific numbers and examples are provided.
- A formula for the coverage is now provided in the metrics section. Further mathematical aspects were added to the Supplementary Material.
- A brief discussion of [M]⁺ and [M]⁻ precursors has been added to the manuscript, along with an additional chapter in the Supplement covering other ionization types.
- The text regarding CASMI 22 data integrity has been reduced in the main manuscript and moved largely to the Supplementary Material.
- UMAP analysis has been shortened, as it is more of a property of GNNs and not specific to the Fiora model.
- All figures were updated to contain the results from MSnLib test split and the improved model.
- The performance of the Fiora-OS (open source) is compared and contrasted with that of the original Fiora model.
- The speed evaluation section has been refactored for clarity, with more specific and detailed results. Hardware specs of the test environment are provided.
- Details in the Methods sections were added, according to the new data and model.
- Data and Code Availability statements have been added to align with the journal's policies.
- A SourceData.xlsx file has been provided containing relevant data for reproducing figures and tables. Scores are labeled with "ice" and "cfm" to denote results from ICEBERG and CFM-ID, respectively. Detailed instructions for generating the outputs can be found in the Jupyter notebooks available on our GitHub.
- Overall, writing throughout the manuscript has been improved for conciseness and clarity.

Please find our point-by-point responses to each comment below, highlighted in blue.

REVIEWER COMMENTS

Reviewer #1 (Remarks to the Author):

Strengths

* This paper presents a neural network to perform single-step bond-breaking for in silico mass spec fragmentation. The effort of improving the efficiency of existing models is a nice effort to make in silico fragmentation models more accessible for the community.

* Considering bond-breaking and predicting two fragments jointly is interesting and sounds novel.

* The experiment study is extensive and detailed.

Weaknesses

* One of the shortcomings of the experiment part is that all models are trained on an older version of the NIST library (NIST'17), given the fact that the baseline method ICEBERG was trained on the newer NIST'20. Are there any justifications for why NIST'17 is considered? It will be a more fair comparison with the baselines if the experiment is performed on NIST'20 (also where ICEBERG is optimized for, in case of any potential overfitting issue on the smaller NIST'17). Also, method developers should always work with the best dataset available to make sure that the downstream user is getting what they expect.

We thank the reviewer for their time and thoughtful feedback. Regarding the use of NIST 17, the simple answer is that it is the only commercial spectral library available at our institute. Therefore, we extended our dataset by aggregating public spectra from MS-Dial to create a joint dataset that is still

comprehensive. This effort has now been doubled by adding yet another open-source library, MSnLib, to the study. Note that retraining ICEBERG is quite complex, as explained below when addressing the speed comparison, so that MSnLib serves primarily as another “hard” test set and facilitates an open-source release of Fiora’s weights, both of which were requested by the other reviewers.

We understand that it is an advantage of ICEBERG to be trained and validated on such a large and new database, but there are a few more things to consider. Goldman et al. make a great concession to include more data by completely excluding collision energies from the evaluation, even though the tools being compared factor these in. As we demonstrate in the Supplement, collision energies are a critical determinant of peak intensities. Furthermore, CFM-ID was never trained on NIST20 either. Our collected dataset, drawn from various sources, aligns more closely with CFM-ID’s in size, offering a fairer comparison across the tools. Goldman et al. never answer the question whether ICEBERG is better than CFM-ID, trained on a comparable dataset and considering collision energies, a gap that our study now answers†. Note that ICEBERG still uses our validation split, preventing overfitting for the most part.

Our training and test data come from a diverse set of sources, with careful preprocessing to maintain the integrity of collision energies, which is quite challenging even for a dataset like NIST17. While the final dataset is smaller than what a commercial library like NIST20 might provide, we believe our approach, leaning more towards open source now, is particularly relevant for methods development in the metabolomics community. Through rigorous statistical analysis and fair discussions, we ensured that ICEBERG is not portrayed negatively.

†Regarding this question, ICEBERG is indeed more accurate than CFM-ID, though this becomes evident only when considering non-precursor fragments, a point now explicitly discussed in the Spectral Prediction Quality section and the Discussion.

*** The authors claimed that the support of both $[M+H]^+$ and $[M-H]^-$ is a novelty. But it is also worth noting that this paper ignored some other common adducts such as $[M+H-H_2O]^+$, $[M]^+$, $[M+Na]^+$, which are considered in other baselines. Is there a specific reason why they are not covered? It seems a natural extension to just add more dimensions to the one-hot encoding of adducts.**

Thank you for pointing this out. Indeed, adducts were not thoroughly discussed in the original manuscript. In response, we have integrated $[M]^+$ and $[M]^-$ ionization types into the new Fiora-OS model, and these are now briefly addressed in the main Discussion. Other adducts are covered in a dedicated section of the Supplementary Material. As explained there, expanding Fiora to include adducts like $[M+Na]^+$ is not straightforward due to the local neighborhood implementation, which is key to Fiora’s predictive accuracy. The manuscript is now clearer about the features Fiora supports and those it does not.

*** The use of single-step bond-breaking, as also discussed by the authors, is probably the major technical contribution and potentially a major bottleneck in performance. It would be better to discuss:**

- What is the exact proportion of peaks that cannot be explained by single-step fragmentation?
- How does the model break rings?
- The authors mentioned that "Fiora implicitly covers multiple bond breaks", is there an example of that?

Indeed, single-step bond breaking presents both an innovation in local neighborhood-based intensity prediction and a potential bottleneck due to the limited set of fragments. Therefore, we have substantially revised the single-step fragmentation section to provide more clarity and include

additional statistics. This revision demonstrates that Fiora's performance remains stable in most cases. To address your three points:

- We added this information for the CASMI datasets. Notably, the peak intensity coverage is higher than the proportion of explained peaks, indicating that our algorithm accounts for high-intensity peaks effectively. For example, in CASMI 2016, 50% of peaks are explained, capturing over 80% of the total ion count. While it is difficult to determine how much better the coverage would be with multi-step fragmentation (as it depends on the comprehensiveness of the implementation), we show across several datasets that ICEBERG and CFM-ID do not perform better, despite having a larger set of fragments.
- The model does not break rings. We have clarified this more explicitly now in the manuscript.
- It is quite challenging to locate a specific example of multi-step fragmentation, as true fragmentation pathways are not encoded in the spectra. However, our algorithm predicts the final ion distribution. Should multiple bond breaks occur on the same residue of the molecule, the model is trained to predict only the final edge break that yields the fragment ion. This means that all intermediate bond breaks on that side of the molecule are already included in the output. Since collision energies are modeled continuously, Fiora can distinguish between breaking smaller parts off the molecule and larger parts that may include (previous) steps of bond breaks, although finding a definitive example to illustrate this is difficult.

*** The introduction of retention time (RT) and collision cross section (CCS) is interesting, but the reported results in this current version seem just proof-of-concept and the accuracy and scalability at the current stage seem not ready to be deployed in real-world tasks.**

To address this, we added a baseline predictor for RT, estimated from logP values. We have also rewritten and shortened this section to make it clear that this has a proof-of-concept status and is not the focus of our study.

*** The clustering effect shown in the embedding of Figure 4(a) is a bit trivial, as has been found in many GNN-based studies of molecules. It seems more like the expressive power of GNN itself, instead of any new improvements brought by learning with the mass spec. It might be interesting to see if there is any correlation between GNN embedding similarities and mass spec similarities.**

In response, we have rewritten and condensed the section, so that it emphasizes the "expressiveness of GNNs" rather than any characteristics specific to Fiora.

We observed a weak positive correlation (Pearson Correlation Coefficient of 0.1) between cosine similarity in the embedding space and spectral cosine similarity. Note that structurally similar compounds can still produce vastly different spectra. The embedding space only accounts for the compound structure graph, without considering key covariates like collision energy. Also note that Fiora – being a fragmentation algorithm and unlike direct prediction models – is detached from the spectral output space. Instead, edge breaks and ion charges are predicted based on local neighborhoods, meaning that even identical fragmentation patterns can result in distinct m/z peaks. As a result, the weak correlation highlights the inherent challenge in metabolomics, where structural similarity does not necessarily translate to spectral similarity. Programs like MS2Deepscore by Huber et al. are specifically trained to bridge this gap, whereas fragmentation algorithms – unfortunately – do not inherently acquire this capability. Nonetheless, it is interesting to examine and discuss such results.

*** It may need more care when comparing the training speed. The authors mentioned that "ICEBERG was trained on only the positive spectra on a GPU for over 6 days", while it is worth noting that**

ICEBERG's code involves many evaluation steps that are not correlated to training. They also run replicate experiments on 4 splits. It needs to be double-checked whether ICEBERG's single "training" step is taking as long as 6 days. Also, it is a good idea to report the CPU/GPU/RAM/etc of the testbed for readers to understand the computational resources needed to run the model.

As suggested by the reviewer, we have added the hardware specs of the test environment to the manuscript.

To be very open regarding the retraining of ICEBERG, in practice it took us over a month to get the software running due to numerous crashes and debugging challenges, largely caused by missing exception handling in ICEBERG. To give only one example, the algorithm could not process element type gold (Au), likely due to an error in the Magma algorithm during step 2 (see Table below), but the crash only occurred during the final training step—after six days of processing.

The reported runtime (6d) reflects a single complete run without errors. For the reviewer's reference, we provide detailed runtime measurements in the table below. We disabled experimental replicas and used the same (random) training/validation/test splits as for Fiora, ensuring the runtime estimate is accurate. Out of the six necessary steps, only two involve actual "training" (one for the generative module and one for the intensity prediction module), and these account for more than half the total time. However, data processing was also considered for Fiora's run time.

Thus, we believe we have been fair in our original statement. However, if required by the reviewer, we can specify that of the six days, approximately three and a half were spent on training, with the rest dedicated to data preparation.

Explicit command	Purpose	Run time
time . data_scripts/all_assign_subform.sh	Data preparation	2045m34,408s
time . data_scripts/dag/run_magma.sh	Data preparation	1351m24,188s
time bash run_scripts/dag_model/01_run_dag_gen_train.sh	Training	1911m46,370s
time python run_scripts/dag_model/02_sweep_gen_thresh.py	Data preparation	0s
time bash run_scripts/dag_model/03_run_dag_gen_predict.sh	Data preparation	296m24,160s
time bash run_scripts/dag_model/04_train_dag_inten.sh	Training	3251m49,794s

Reviewer #2 (Remarks to the Author):

The authors present a novel approach for the prediction of high-resolution tandem mass spectra, based on a graph neural network approach. The approach is based on predicting intensities for all single fragmentation events arising from a single fragmentation event. The paper is well written and to the point. The code is freely available. The authors openly discuss strengths and are honest about the limitations (single fragmentation) of their approach. Collision energy as a continuous input value is a nice addition over many existing models. In summary, nice work and certainly worth publication.

My only serious criticism is about the choice of not excluding precursor peaks from spectra for

evaluation. The authors argue that:

"Other adjustments to the cosine score include removing the precursor [... This] may eliminate a dominant precursor peak in some cases, but significantly increases the overall cosine bias. Second-order dominant peaks may arise from obvious losses, such as H₂O, and the molecular stability and the relationship of the intact molecule to the fragments are a indispensable parts of the fragmentation pattern. This makes precursor removal rather futile for spectra that already contain very few peaks [...]"

I strongly argue that the authors should add a second evaluation with removed precursor. The unfragmented precursor mass is not structure-specific for a candidate of the same formula, and it is common practice to exclude precursor from spectral library search for that reason. (A small neutral loss, such as H₂O, at least indicates something about the structure). For this, it doesn't matter if the spectrum only has few peaks. It is obvious that not excluding precursor inflates the cosine score for low collision energies (as is easily seen in Supplementary Fig 3 and 4). We care less about cosine bias than about seeing whether the model can solve the hard task, namely predicting the fragmentations. If a model can show surface-level strong performance numbers by predicting unfragmented precursor intensity (which would be excluded in search anyway), this obfuscates the real performance of the model for the cases we really care about and that really make a difference. I don't care about the cosine being nominal 0.8 or 0.7, as this is such a dataset-dependent number, I care about the performance on a **hard** problem compared to baselines and SOTA.

We sincerely appreciate the reviewer's detailed input on this matter. In response, we have conducted a full second evaluation of model performance excluding the precursor peak. Fiora still outperforms the other tools in 7 out of 8 cases (with the exception being CASMI 22 positive spectra). However, as expected, the margin of improvement is smaller compared to the full-spectrum evaluation, given that Fiora benefits from accurately modeling the precursor peak in relation to collision energies. This is thoroughly discussed in the Results section. Additionally, we have overhauled the Metrics section (in the Methods chapter), as the previous version admittedly oversimplified the impact of precursor removal. An additional chapter in the Supplementary Material delves further into the distribution of cosine similarity, offering statistical insights and expanded comparisons that, we hope, convinces the reviewer of Fiora's superior performance over the other tools.

(As we know, model evaluations should be such that they are "hard but not too hard" for a model. Böcker 2017: "In summary, a challenge must not be too hard, or all methods will fail. But if you evaluate a method in its 'comfort zone', there is not much to be learned. If you want to see how strong a piece of concrete is, you have to break it.")

Other than this, I have only few comments:

* General: Particularly driven by the interpretation Fig 4b, it would be good if (all) box-whisker plots had the number of class instances (n=??) for every bar.

We have added the number of unique compounds for each group in Fig. 3 and Fig. 4, displayed as histograms above the box-whisker plots. In Fig. 2, all subsets are evaluated using the entire validation split, resulting in an equal number of compounds in each group.

* p8

"Still, the relative loss of performance loss is lower..."

I wouldn't read too much into this honestly, as the magnitudes are small.

This sentence has been removed, and the discussion now focuses more on the broader interpretation of the findings.

* P9, 10 RT/CCS: As presented, it is a bit of a gimmick feature. It would be more substantial if compared to some baselines, at least very simple ones such as logP for RT and pure m/z predictor for CCS, to show that it's better than just throwing darts or drawing a rough line. Preferably should compare to models closer to the state of the art.

To address this, we used the xlogp3 program to calculate logP values for all compounds and set up a linear regression (LR) estimator for RT, for a baseline comparison. CCS values were similarly compared to LR based on precursor m/z. In all cases, Fiora's predictions outperformed the baseline LR. However, this remains in a proof-of-concept stage, as a thorough evaluation of RT and CCS prediction would require a dedicated study.

* P11, comment on CASMI22:

Obviously these spectra are somewhat strange. The cynical reading of this is, however, that FIORA is good at low CE because it predicts the unspecific precursor, and bad at high CE because it misses multiple fragmentations. Just another reason to add a validation with precursors removed. It is still a consolation that everyone is bad at CASMI22.

With the precursor exclusion in place now, we observe that scores do not change much regarding CASMI 22. In fact, this dataset is least affected by the change. Notably, predictions in high energy setting (65% NCE) continue to show worse performance across all tools, as detailed in Supplementary Figures 12 and 13.

However, we can make a few interesting observations in relation to CE. Supplementary Figure 10 demonstrates that prediction performance, without the precursor, remains stable across most energy levels and only declines significantly above 60eV. We interpret the following:

1) Fiora indeed benefits from more correctly estimating precursor intensity, as low-energy scores are consistently higher with the precursor, but not so without. This is also highlighted by the example predictions, shown in Supplementary Figures 5-7, where Fiora accurately estimates precursor intensities and their degradation when CE increases. Having this context makes interpretation of the spectral prediction more plausible, and not integrating CE – like in ICEBERG's algorithm – is an overall loss of information, even though precursor m/z is unspecific to the structure.

2) The impact of missing peaks appears to become significant primarily at very high CE (above 60 eV). Note that we also observe reduced performance for ICEBERG and CFM-ID, indicating that multi-step fragmentation does not fully address the issue. High-energy settings seem to present inherent challenges for spectral prediction. This may also be due to the underrepresentation of high energy spectra (seen in Supplementary Figure 14).

* P17, Table 3: I commend the authors for the diligent filtering. Kind of sad numbers in the end, compared to the nominal size of those libraries. (Makes me wonder whether the MS-Dial includes the

up-to-date GNPS and MassBank. Maybe try to get the cleaned-up current GNPS by Huber/deJonge and the current MassBank 2024.06 release. Still, it's clear that NIST will give better results anyway.) I would be interested (maybe in the SI) in a step by step reduction of cpds / spectra after each filtering step.

Likely, MS-Dial does not contain the most recent data (and covers only a small fraction of GNPS), but it includes proper annotation for collision energies, as well as CCS and RT values. To address this shortcoming, we opted to incorporate the new MSnLib by Brungs et al., which offers a large number of open-source spectra in a well-formatted manner. While GNPS-v2 (cleaned up) does contain some CE data, the units (e.g., NCE vs CE) are unclear, and it likely overlaps with MSnLib. MSnLib, therefore, is a suitable fit for training Fiora's open-source version while also providing a reliable test set.

Regarding the step-by-step reduction, we direct you to our public Jupyter notebooks, which offer an accessible overview, plots, statistics, and the impact of two rounds of filters. For MSnLib see: https://github.com/BAMeScience/fiora/blob/main/lib_loader/msnlib_loader.ipynb

Note that it is a bit tricky to provide exact numbers for each step as it depends on the order in which filters are applied, so we chose not to include this information in the already extensive manuscript. All processing steps, as well as training and test scripts used to generate the figures in the manuscript, are documented in the Jupyter notebooks, which can be examined without running the code.

*** P18**

“Placing additional emphasis on higher ion masses, as suggested by Stein & Scott (1994) [51], for instance, by multiplying peak intensities by m/z values, makes sense in order to increase the score for more compound-specific fragments. Fragments ions with low masses are more likely to be shared between different compounds.”

This adjustment was developed for GC-MS data, where the precursor mass is not known. It cannot be transferred to the ESI-MS/MS case where the largest mass (isolated precursor) is actually the least specific by definition, and other large masses (like neutral losses) are also relatively unspecific, as the authors themselves note above. (Not that it matters much, since the authors didn't include such an adjustment.)

Thank you for pointing this out; it escaped our notice. With the additional space now dedicated to the precursor peak discussion, we have decided to remove this section from the manuscript.

Reviewer #2 (Remarks on code availability):

I haven't reviewed the code in detail, but gave it a quick glance and made sure I can install and run it. I tested the version from the preprint, which installed fine, but there are no weights included because they rely on closed-source data, so one would have to retrain or ask the authors.

We now provide open-source model weights, trained on MSnLib data, which are ready for use upon installation. Users can easily test the model by following the example command in the Readme. For a hands-on experience with sample predictions (and spectral plots), we recommend the `live_predict.ipynb` Jupyter notebook (available at

https://github.com/BAMeScience/fiora/blob/main/notebooks/live_predict.ipynb).

Reviewer #3 (Remarks to the Author):

This is an interesting study. I have always wanted to implement a similar approach, so I enjoyed (in part) learning from the results of this study. It is very commendable that the authors have compared their algorithm with a yet-to-be-published tool reported in a preprint. I appreciate that the authors want to demonstrate the ability of FIORA to be used with RT and CCS, and I also understand that they want to play that part down since it is not the main paper focus. However, the paper is 'obscure', meaning that has parts that are hard to understand, that it has some incongruences, and that some parts are described in a highly technical way, not suitable for the wide audience where articles in Nat Comms are written for. Generally, the write-up could be condensed to the important points. It could also benefit from additional results and metrics. I recommend the authors take a look at the following paper: [10.1021/acs.analchem.2c02093](https://doi.org/10.1021/acs.analchem.2c02093), which uses intuitive metrics to assess and compare the performance of the prediction algorithm, including the top k performance (a metric used in ICEBERG as well).

We thank the reviewer for their thoughtful feedback and expertise. We have thoroughly revised the Results, Discussion, and parts of the Methods to enhance clarity and conciseness. Sections with excessive detail, such as the CASMI 22 discussion, were moved to the Supplementary Material. We also introduced several new statistics and generated figures akin to those in the recommended paper by Zhu et al., as detailed in our responses below.

While we agree that a top-k compound retrieval metric is a valuable addition, setting up a spectral library search pipeline was not feasible within this revision due to the numerous changes and other additions made. Instead, we tested cosine similarity without the precursor peak, providing further insights into the models' ability to distinguish compounds with identical precursor weights. This should serve as a reasonable proxy for the top-k retrieval recall rate.

An entirely new section discussing alternative scoring functions and cosine similarity distributions was added to the Supplementary Material. Additionally, we incorporated a new test dataset using MSnLib, which nearly doubles the number of compounds for validation. This allows us to demonstrate Fiora's prediction quality across various metrics and test cases, while keeping the main manuscript focused and concise.

Some of the 'obscure' things that I observed:

- From what is understood from the main text, the authors report the main algorithm results and comparisons with the cosine similarity. At the very end of the paper, in the methods section, the authors then state that they use the square root transformed peaks, something that the authors claim is common in metabolomics (without a reference).

We added a reference to the statement. It is common in spectral search software, such as *mzmine*, to use square root transformed intensities by default, so most reported cosine scores already incorporate this adjustment.

- The authors mention that they have "retrained" (literally) the ICEBERG algorithm with the same dataset used for FIORA, but then they contradict themselves. It seems that the authors use the metrics reported from the preprint of ICEBERG, but ICEBERG seems to be trained with NIST20 instead of NIST17.

ICEBERG was retrained, and all metrics presented are based on this retraining. However, we compared our findings with those of Goldmann et al. in the ICEBERG study. To avoid confusion, we removed that section and clarified the specific data used for comparison.

- Fig 2 and Fig 3, Fig 2 says that the vertical lines correspond to the 95% CI. For Fig 3 this is not defined, and I assume that these are the same. If that is correct, what these figures show is very hard to believe, especially from what is then observed in Fig 6. It implies the algorithm has incredible precision. Please report the accurate 95% CI or at least the IQR.

Please note that confidence intervals, representing the uncertainty of the median estimation, are accurately calculated and have been explicitly verified with the code below. However, it seems the main consideration here is whether the focus should be on displaying uncertainty or emphasizing the overall distribution spread. The spread, measured through standard deviation and IQR, is quite large for all predictions, whereas confidence intervals are smaller due to the large number of data points.

We have carefully reviewed all error metrics and, given the large variance that compromises the comparisons, concluded that the 95% confidence interval (CI) is more appropriate. It reflects the reproducibility of mean/median results, which serve as model selection metric on the validation set and the comparison of performance across the three software tools. The Figures below illustrate the differences between 95%CI and IQR (50% percentile interval) estimates. In Supporting Fig. 2, validation performance cannot be properly compared due to the high variance. Note that standard deviation is similar to the IQR. Goldman et al., in the ICEBERG study, also presented results with 95% CI (calculated by 1.96 times the standard error). Further statistics on the cosine score distributions are provided in the Supplementary Material, clearly demonstrating that Fiora and ICEBERG follow comparable distributions, with Fiora consistently achieving higher cosine scores. Accordingly, we believe that, as in the ICEBERG study, the 95% CI is the more appropriate error metric for comparison. The manuscript has been updated to clarify the error metrics used and their relation to standard deviation.

Supporting Fig. 1: Using 95% CI of median estimation

Supporting Fig. 2: Using IQR (50% Percentile Interval)

Verifying that confidence intervals are correctly calculated.

The in-built Seaborn function for generating the plots is:

```
sns.pointplot(data=L, x="depth", y="validation", estimator="median", capsiz=0, markers="o",
palette=color_blind4, markersize=5, errorbar=('ci', 95), linestyle='--', hue="gnn_type", dodge=0.4)
```

We verified the displayed result by calculating the confidence intervals with SciPy from both the t-distribution and bootstrapping, confirming the accuracy of the CI shown in the figure. You can review the code below and also on GitHub:

(https://github.com/BAMeScience/fiora/blob/dev/notebooks/grid_stats.ipynb).

```
import numpy as np
import scipy.stats as st
from scipy.stats import bootstrap

def print_stats(scores):
    print(f"Num of spectra: {len(scores)}")
    print(f"Median: \t{np.median(scores):.3f}")
    print(f"Var: \t{np.var(scores):.3f} (Standard deviation: {np.std(scores):.3f})")
    conf_in_t = st.t.interval(confidence=0.95, df=len(scores)-1, loc=np.median(scores), scale=st.sem(scores))
    conf_in_boot = bootstrap((scores,), np.median, confidence_level=0.95)
    print(f"95%CI: {conf_in_t} (from t distribution)")
    print(f"95%CI: {conf_in_boot.confidence_interval} (from bootstrapping)")
```

```
val_scores = L[(L["gnn_type"] == "RGCN") & (L["depth"] == 6)]["validation"]
print_stats(list(val_scores))
```

```
Num of spectra: 7212
Median: 0.821
Var: 0.042 (Standard deviation: 0.206)
95%CI: (0.8160764988782188, 0.8255678083244333) (from t distribution)
95%CI: ConfidenceInterval(low=0.8160298742214132, high=0.825839917098286) (from bootstrapping)
```

For example, in Supporting Fig. 1 (and in the manuscript), the median RGCN at depth 6 is 0.821, with a 95% confidence interval of [0.816, 0.826].

- "Out-of-distribution bias": a complicated way of saying lack of generalization (a term later used by the authors).

In response, we rephrased the sentence.

- The whole section about speed is not clear: it is faster only on the GPU? Table 2 is misleading, as it seems that a CPU is not faster, and it is not clear if Table 2 results are CPU or GPU.

To address this, we added a column in Table 2 to specify device types (CPU/GPU), and the text was revised to clearly state that performance improvements depend solely on GPU utilization.

- The section's impact on single-step fragmentation is also cryptic and hard to follow. E.g., how is the dotted line calculated?

In response, we have completely revised the single-step fragmentation section for enhanced clarity, providing examples that facilitate reader understanding. We also mathematically defined the term "coverage" in the Methods section. Furthermore, we included a Mathematical Aspects section in the Supplementary Material that details how to derive the maximum cosine similarity for a given coverage (represented by the dotted line).

- The manuscript does not have paragraphs or page lines, which hampered the job of this reviewer.

We apologize for the inconvenience. To enhance readability, line numbers, increased paragraph spacing, and wider page margins have been added.

- What is more surprising is to read sentences in the results section like: 'overall stellar performance', 'it is all more impressive', 'it is remarkable'.

All these instances have been deleted to maintain scientific integrity.

Other concerns that I have:

1. Collision energy (CE) is not included or discussed in the results, especially given the singularity of FIORA using the one-atom break strategy, and given also the importance of CE in MS/MS comparison in metabolomics. First, it is not clear at all which CE does the algorithm cover. Second, I would like to see how FIORA fails at predicting low-intensity peaks as the CE increases. Overall, some mirror plots for the prediction, similar to the ones used for CMF-ID, ICEBERG, or in 10.1021/acs.analchem.2c02093, could improve the results section.

In response, we added a brief discussion on collision energies to the manuscript and an in-depth chapter to the Supplementary Material. Fiora accommodates collision energies ranging from 0 to 100eV as continuous input values, which is now explicitly mentioned in the manuscript. Additionally, we produced stacked mirror plots like those in the suggested study. These highlight an average prediction example, closely reflecting the median performance of all tools. In the Supplement, the impact of collision energies on the same compound is illustrated using the mirror plots, highlighting Fiora's precise portrayal of precursor intensity but reduced accuracy in predicting smaller fragment ions at higher collision energy.

2. I appreciate the results from Fig 3, but the authors could use the GNPS library to demonstrate the generalization capability, as it is also quite different from the data contained in NIST/MS-DIAL and is much larger than CASMI and widely used.

This is a valuable suggestion. To further assess generalizability, we incorporated the newly released MSnLib, which serves a similar role to GNPS as described by the reviewer. MSnLib is notably easier to manage concerning metadata like collision energies. In the Supplement, we demonstrate that most of the structurally dissimilar compounds come from MSnLib, making it an ideal test scenario. This data has also been added to Fig. 3, along with the compound count for each similarity group, giving it more statistical significance.

3. Fig 6 could be plotted with the results from the validation set, as it makes more sense given the high dissimilarity between the CASMI data and the training data.

Validation and Test spectra were intentionally excluded due to the overwhelming number of data points, which would be difficult to represent in a parity plot. However, the two CASMI datasets offer clear examples of both high and low coverage scenarios. This is now explicitly discussed in the section. The relation between structural similarity and cosine scores is further explored in the Supplementary Material.

4. The results for the comparisons made against ICEBERG and CFM-ID should be shown with more statistical rigor: cosine similarity boxplots for the three algorithms. I understand the rationale behind the use of the root square cosine, but it would be interesting to see the comparison made with plain cosines (unweighted). What about using other metrics like entropy (10.1038/s41592-021-01331-z)?

We have dedicated an entirely new section in the Supplementary Material that explores the distribution of cosine similarity and its variations, such as unweighted (raw) cosine similarity and cosine similarity excluding the precursor peak. This section includes histograms and kdeplots to visualize the distribution of cosine scores, and boxplots that capture the impact of varying collision energies across all tools. We find that Fiora exhibits very similar distributions to ICEBERG with consistently higher scores. This reaffirms Fiora's high prediction quality across several metrics and datasets, strengthening the statistical credibility of our findings.

Please note that we believe a table is well suited for the main manuscript, as it portrays clear performance trend across 8 test cases, given that the manuscript already contains numerous figures. All the while, additional statistical analyses are found in the Supplement.

5. It is not described in the methods section (Fig 7) how ambiguities are solved: e.g., in Fig 7, one experimental peak could correspond to the formula C₄NO₁ (without considering the H). This formula could come from two possible substructures (i.e., there are two different groups of nodes whose atoms could correspond to C₄NO₁). How do the authors know from which of the two possible substructures this peak stems from?

Thank you for bringing this up. This aspect was missing from the Methods section, but we have now clarified it. In cases where multiple fragments can explain the same peak, they are assumed to contribute equally. While this is a simplified assumption, it works effectively as such conflicts are rare in single-step fragmentation. For spectral prediction, peak intensities are summed across all fragments producing the same peak.

Regarding your example, the lack of multi-fragmentation prevents the C₄NO fragment from being considered. However, if it were, with two possible fragmentation pathways leading up to it, equal probability of edge breaks would be assigned during training derived from the ground truth peak intensity matching the m/z values. During prediction, the intensities of both fragments would be summed into a single peak. Fragments are only merged if they share the same structure; different structures with the same formula produce two separate peaks at the same position. This distinction is mainly relevant for structure annotation, as peak intensities are summed for scoring and plotting regardless.

6. Data preparation: the authors state "converting NCE to eV". How did the authors approach this? Why such a large ppm error (50 ppm) was used? "90% of precursor intensity" was used at most? I do not understand it. But if I guess correctly, this would limit the inclusion of spectra acquired at high CE, as higher CE yields smaller precursors.

Thank you for the input. We clarified the following in the manuscript:

- The conversion of CE formats follows the Thermo Fisher formula from the frequently cited proteomics news blogpost.
- The high ppm threshold is due to uncertain machine accuracy. Empirically, we found that using thresholds below 50 ppm caused correct peaks-fragment matches to be missed, particularly for MS-Dial spectra. The new MSnLib uses a 10 ppm error, and this choice is now explained in the Methods section.
- The precursor maximum of 90% restricts the inclusion of low CE data, ensuring that at least 10% fragmentation occurs. This ensures that the models learn fragmentation patterns beyond merely predicting the precursor peak.

7. The trained model object does not seem to be published in the Github repository, hampering the wide adoption of this tool by the community.

We are pleased to announce the release of fully open-source model weights, trained on MSnLib, now available on our GitHub. These weights are automatically selected upon installation for ease of use.

Response Letter

Dear Editor, dear Reviewers,

We sincerely thank the reviewers for their thorough evaluation and constructive feedback on our manuscript. To address the remaining minor points, we have added further supporting data to the study, including top-k recall performances, and prepared a detailed point-by-point response.

In response to the editor's suggestion to include the NIST'20 library or to provide a clear explanation of its potential impact on our results, we would like to respectfully clarify our decision. After careful consideration, we concluded that incorporating this library in place of NIST'17, while requiring considerable time and cost, would not significantly alter our findings or enhance the paper. This is primarily due to the comprehensive compound coverage already achieved by the newer, open-access libraries included in our study, which are actively updated. Importantly, our approach prioritizes accessibility and reproducibility, allowing researchers without access to library licenses to adopt and extend our methods –a flexibility not supported by NIST's proprietary, closed-source format. A detailed explanation of our reasoning is provided in the comments below.

We hope these revisions satisfactorily address the reviewers' comments. Please do not hesitate to reach out if further clarifications are needed.

Kind regards, on behalf of all authors,

Yannek Nowatzky

Please find our point-by-point responses to each comment below, highlighted in blue.

REVIEWER COMMENTS

Reviewer #1 (Remarks to the Author):

I really appreciate the authors' response and the thorough update to the manuscript. After reviewing the revised submission and reading the comments from other reviewers, I still have a few remaining questions and concerns:

We sincerely thank the reviewer for their thorough and valuable feedback.

* I understand that the authors used NIST'17 due to licensing constraints, and I appreciate the clarification. However, I just wanted to note that not using the latest dataset might impact the perceived technical novelty of the work. For anyone who may not be familiar, NIST licenses typically cost around \$2,000.

We understand the concern but respectfully argue that this study should remain concluded with NIST'17. In addition to NIST'17, we have incorporated two other libraries: MSDial (last updated 8/2022) and the newly released MSnLib (last updated 10/2024). By the time of writing, MSnLib already matches NIST'20 in compound count and continues to grow rapidly, with five open-source updates released in recent months (refer to the DOI and zenodo-link). In contrast, NIST has removed export functions to common spectral format, such as MSP format, in its '23 edition, accompanied by a statement of intent indicating that methods development or benchmarking was not its primary purpose (see webpage). Thus, the technical novelty lies with the new open-source libraries we have already incorporated.

Incorporating NIST'20 would entail additional costs and significant effort, without a reasonable expectation of altering the results. Fiora has demonstrated high consistency across diverse conditions, such as Tanimoto distance, compound classes, ionization types, and covariates (refer to Figs. 3, 4c, and Suppl. Figs. 3, 4, 8-13, 16, 20, 21), while displaying highly similar distributions to ICEBERG. Furthermore, our newly introduced top-k retrieval performance already surpasses ICEBERG's reported results on NIST'20 (Goldman et al., 2024). Considering all the above, we have no reason to believe that using NIST'20 would yield meaningful differences in outcomes.

In summary, the combined data used in this study already outpaces NIST'20 in terms of compound counts, experimental conditions covered, metadata richness, and novelty of release. By focusing on publicly available and regularly updated resources in addition to the well-established NIST'17 library, we aimed to ensure reproducibility and accessibility, enabling researchers without licensing budgets to engage with our methods and software. We believe this combination balances the practical and innovative aspects of our work effectively.

* In my initial review, I didn't notice that the median is used instead of the mean. Could you clarify the reasoning behind this choice?

The reason is that scores are not normally distributed but show a skewed distribution (see Supplementary Fig. 3). Therefore, we believe that the median better represents the distribution of MS/MS predictions and their cosine scores. Ultimately, this choice does not affect the performance comparisons. As shown in the screenshots below, the relative differences between tools remain consistent regardless of the metric.

Summary test sets

```
Out[139]:
```

	model	Test+	Test-	MSnLib+	MSnLib-	CASMI16+	CASMI16-	CASMI22+	CASMI22-
0	Fiora	0.766878	0.746718	0.631179	0.584562	0.696171	0.704716	0.342100	0.340091
1	CFM-ID 4.4.7	0.624463	0.548172	0.482897	0.424296	0.682091	0.550478	0.396155	0.342921
2	ICEBERG	0.676692	NaN	0.545627	NaN	0.677182	NaN	0.381035	0.000000

Supporting Fig. 1: Mean performances across all test sets

Summary test sets

```
Out[140]:
```

	model	Test+	Test-	MSnLib+	MSnLib-	CASMI16+	CASMI16-	CASMI22+	CASMI22-
0	Fiora	0.807246	0.791260	0.645026	0.610936	0.766605	0.773664	0.289139	0.317697
1	CFM-ID 4.4.7	0.666676	0.565989	0.482090	0.409999	0.704946	0.589614	0.376012	0.292884
2	ICEBERG	0.716473	NaN	0.575536	NaN	0.709359	NaN	0.358571	0.000000

Supporting Fig. 2: Median performances across all test sets

* The claim that “Fiora implicitly covers multiple bond breaks from the same residue” is a bit unclear to me. When reading the manuscript, I interpreted this as “Fiora has the ability to break multiple bonds beyond single-bond breaking,” but the authors have confirmed this isn't the case.

We have rephrased the sentence to “indirectly covers”, which indicates more clearly that Fiora's algorithm does not explicitly break multiple bonds at a time.

Line 359: It should be noted that Fiora **indirectly** covers multiple bond breaks from the same residue, as is explained in the Fragmentation algorithm section.

* In your response to Reviewer #3, I noticed the requested top-k retrieval study wasn't included. Instead, you mentioned that cosine similarity (without the precursor peak) serves as a reasonable proxy for the top-k retrieval recall rate. I respectfully differ in opinion here; the top-k retrieval study is a well-recognized metric in real-world applications of your model and has been utilized in both the ICEBERG and CFM-ID papers. It offers a more direct and convincing assessment than cosine similarity.

In response, we implemented top-k recall statistics following the methodology of the ICEBERG study, based on 50 candidates per spectrum retrieved from PubChem. Due to the computational intensity of this process, we limited the comparison to ICEBERG and Fiora, evaluated for both cosine similarity variants on the test split. The results, presented in Supplementary Figs. 20 and 21, demonstrate consistent performance improvements aligned with the differences in median cosine similarity. Incidentally, Supplementary Fig. 22 suggests that precursor removal benefits compound retrieval, though variants incorporating the

precursor might enhance retrieval performance even further. We have updated the manuscript to reference these findings (Lines 240 - 244).

Additionally, I want to make a note on the training speed of ICEBERG — not as a “weakness,” but just as a point of comparison. I’ve run their open-source code myself, and while I don’t recall the exact training times, I remember both scripts finishing significantly faster than six days. Also, the ICEBERG paper mentions, “Models are trained on a single RTX A5000 NVIDIA GPU (CUDA Version 11.6) in under 3 hours for each module.” It may sound like a simple question, but did you double-check that GPU resources were fully utilized for training? I know deep learning model setups can be challenging.

We ran the pipeline on an A100 NVIDIA GPU and could monitor active GPU usage. However, this did not exceed more than 25% of the GPU’s computing capacity at any given time. As to why this is the case, we cannot offer a definitive explanation.

Reviewer #2 (Remarks to the Author):

The authors now included the important analyses without precursor peak, thank you! Also great to have open weights. No further comments.

We thank the reviewer for their constructive feedback, which has been crucial in shaping the current state of the manuscript.

Reviewer #3 (Remarks to the Author):

I thank the authors for their thorough revision and for answering my concerns. I believe the manuscript has been greatly improved. I only have a minor comment:

Regarding the CI intervals. Thank you for the answer, indeed, the issue here is that the CI is low due to the large number of data points. I strongly believe that providing the IQR is a more insightful alternative. Although I understand that authors want to show the best possible side of their results, it is fine to have a large variation, and it gives the readers a more transparent view regarding the expected error range of their predictions. At least, include the IQR range as a supplementary figure (and referenced in the manuscript). This applies to Figure 2, but especially 3. Regarding the best way to present the results, I find supplementary figures 8 and 10 the most convincing in showing Fiora's superiority to the other models.

The same applies to SI Figure 16, I would like to see the IQR for this one as well. The CI shows that, comparatively, FIORA is superior, but I am interested in knowing what variation I can expect in my predictions if I use FIORA.

Thank you for the positive and constructive feedback. Your comment has convinced us to change the error metric to the IQR for Fig. 3 and SI Fig. 16 (Tanimoto scores), as it better captures the variance in cosine scores.

For Fig. 2, however, we chose to retain the 95% CI for better visual clarity. The high variance in this case is not the focus of interest and obscures the subtle differences in the performances of the backend graph architectures. To address this, we have included the corresponding SI Fig. 23 with the IQR in the Supplementary Material and remarked on the difference in variance in the manuscript.

Jan 30th, 2025

Response Letter

Dear Editor, dear Reviewers,

We sincerely thank you for your continued evaluation and constructive feedback on our manuscript. To address the final remaining point regarding our use of the NIST'17 library, we have incorporated a dedicated discussion into the manuscript. You will find that we have thoroughly revised the Data section to clarify the implications of our choice, standard testing conventions, and why this decision does not detract from the quality of our algorithm or the paper. Further details can be found in our point-by-point response below.

We believe these revisions fully address the reviewers' comments while completing all editorial requirements.

Kind regards, on behalf of all authors,

Yannek Nowatzky

Please find our point-by-point responses to each comment below, highlighted in blue.

REVIEWERS' COMMENTS

Reviewer #1 (Remarks to the Author):

Thank you for your response and the updates. My evaluation of the manuscript at its current stage is as follows:

- 1) The manuscript presents technical novelty that I believe will be of interest to the community.
- 2) However, as a deep learning paper, I think it is important to include performance metrics that have been used by recent peer methods. This aligns with the points I raised in my comments and the authors' response to NIST'20. The proposed MSnLib is great, of course, but there seem no previous methods being evaluated on that.

In response, we have thoroughly revised the Data section (Line 157 in tracked changes PDF) to directly address this concern. We now explicitly outline the limitations of the current library (compared to NIST'20) and explain why our evaluation remains on par with the state of the art. Notably, we have tested a broad range of conditions across four fundamentally different test sets, supported by 32 figures and seven tables, to ensure a level of generalizability and rigor that exceeds comparable studies.

The manuscript now discusses the unconventional aspects of our evaluation while highlighting the benefits of incorporating multiple data sources—reducing homogeneity and increasing diversity—as well as our commitment to open-source accessibility. We believe MSnLib will play a key role in shaping future evaluation standards and that this work serves as an important step toward advancing the field.